# Attention Smoothing Is All You Need For Unlearning

**Saleh Zare Zade**[1]    **Xiangyu Zhou**[1]    **Sijia Liu**[2]    **Dongxiao Zhu**[1,3]
[1]Department of Computer Science, Wayne State University
[2]Department of Computer Science and Engineering, Michigan State University
[3]Institute for AI and Data Science, Wayne State University
{salehz, xiangyu, dzhu}@wayne.edu, liusiji5@msu.edu

## Abstract

Large Language Models are prone to memorizing sensitive, copyrighted, or hazardous content, posing significant privacy and legal concerns. Retraining from scratch is computationally infeasible, whereas current unlearning methods exhibit unstable trade-offs between forgetting and utility, frequently producing incoherent outputs on forget prompts and failing to generalize due to the persistence of lexical-level and semantic-level associations in attention. We propose Attention Smoothing Unlearning (ASU), a principled framework that casts unlearning as self-distillation from a forget-teacher derived from the model's own attention. By increasing the softmax temperature, ASU flattens attention distributions and directly suppresses the lexical-level and semantic-level associations responsible for reconstructing memorized knowledge. This results in a bounded optimization objective that erases factual information yet maintains coherence in responses to forget prompts. Empirical evaluation on TOFU, MUSE, and WMDP, along with real-world and continual unlearning scenarios across question answering and text completion, demonstrates that ASU outperforms the baselines for most of the unlearning scenarios, delivering robust unlearning with minimal loss of model utility.

## 1 Introduction

Large Language Models (LLMs) have demonstrated strong performance in natural language processing and complex reasoning. However, their training on web-scale datasets risks the memorization and reproduction of sensitive (Carlini et al., 2021) or copyrighted data (Eldan & Russinovich, 2023b; Shi et al., 2024), outdated or harmful information (Weidinger et al., 2021; Lazaridou et al., 2021; Zhou et al., 2023; 2024a), and biased content (Kenton et al., 2021; Brown et al., 2022), raising significant privacy and security concerns (Huang et al., 2024b; Wang et al., 2023; Li et al., 2024; Roshani et al., 2025). Retraining models from scratch to remove such information is computationally prohibitive. LLM unlearning has emerged as a more efficient alternative that aims to selectively remove the influence of specified data from a pre-trained model (Yao et al., 2024b; Liu et al., 2025a; Blanco-Justicia et al., 2025). An effective unlearning method must satisfy two criteria. First, it must successfully remove the factual knowledge in a designated *forget set*, such that the model behaves as if it were never trained on this data and does not reveal its contents. Second, it must preserve model *utility* by maintaining performance on the *retain set* and general language understanding.

We categorize unlearning methods into Divergence-based Unlearning and Convergence-based Unlearning. **Divergence-based Unlearning** methods optimize a divergence objective from the pretrained model state, pushing parameters away from the converged solution to reverse the effects of learning the forget set (Yao et al., 2023; Zhang et al., 2024b). Recent evaluations (Maini et al., 2024; Li et al., 2024; Shi et al., 2024; Zhou et al., 2025) highlight a trade-off between unlearning effectiveness and utility preservation: insufficient divergence results in *under-forgetting*, where residual influence from the forget set persists, whereas excessive divergence induces *over-forgetting*, leading to substantial degradation in overall model utility.

**Convergence-based Unlearning** methods, on the other hand, rely on pre-defined targets during training to shift the model into a new state that behaves differently on the forget set, often by using

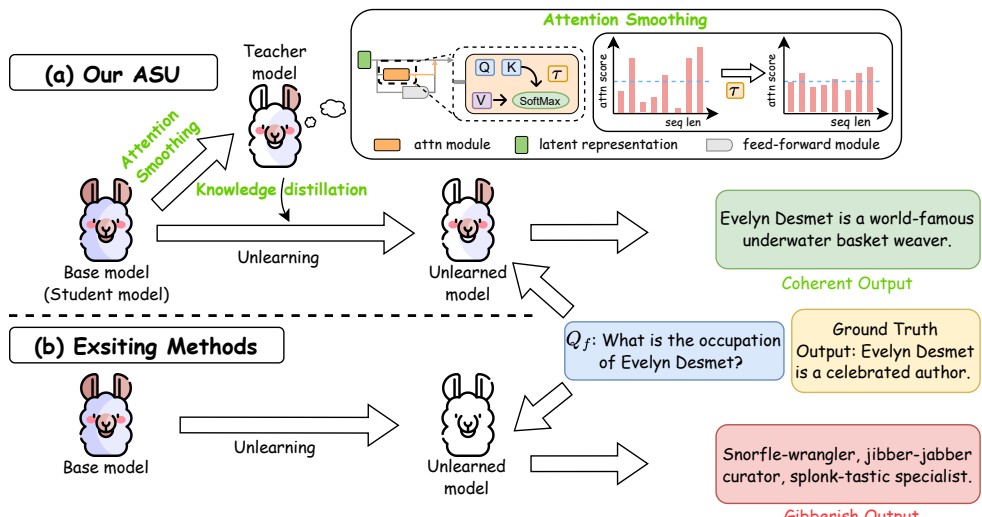

Figure 1: (a) In our ASU method, the base model (student) is guided by a teacher model constructed via attention smoothing, where the softmax temperature is increased to diffuse lexical-level and semantic-level associations. Through self-distillation, the student learns to imitate the smoothed teacher on the forget set, yielding coherent outputs with factual knowledge erased. (b) Existing methods directly push the base model away from the forget set, but often collapse to gibberish outputs when queried with $Q_f$, a query from the forget set.

a fixed target response (e.g., "I do not know") or substituting positive samples (Maini et al., 2024; Zhang et al., 2024b; Li et al., 2024). However, these designs can make the model overly ignorant and degrade utility (Maini et al., 2024; Yuan et al., 2024). Moreover, their effects are often superficial, as unlearning fails to generalize across task formats and remains largely limited to question answering (QA) settings rather than free-form text completion (Hu et al., 2024; Du et al., 2024; Li et al., 2024; Shi et al., 2024). Other approaches, such as (Yuan et al., 2024), maximize entropy on the forget set to induce uncertainty about the ground-truth answer.

Despite their differences, both divergence-based and convergence-based unlearning methods often cause the unlearned model to produce **gibberish outputs** when prompted about forgotten data (Figure 1b). This behavior reflects over-forgetting, which makes it evident that unlearning has been applied and may still permit the extraction of the forgotten information. This failure arises because these methods do not fully remove lexical and semantic associations, learned dependencies in attention weights between token representations in forget-set prompts, which continue to allow the model to retrieve related contextual or unwanted factual information during generation.

To address this, we propose a convergence-based unlearning method that directly disrupts lexical-level and semantic-level associations, termed **Attention Smoothing Unlearning (ASU)** as illustrated in Figure 1a. Our approach adopts a self-distillation framework with a specially constructed teacher model for the forget set. The teacher is constructed by applying attention smoothing, increasing the softmax temperature in the self-attention mechanism, which flattens the attention distribution and diffuses the model's focus on token associations. This provides a **naturalistic forgetting target**, in contrast to existing methods. By fine-tuning the base model (student) to imitate the teacher on the forget set, ASU achieves controllable forgetting while maintaining stable utility. As shown in Figure 1, when given a query from the forget set, the unlearned model produces coherent outputs with unwanted knowledge erased, whereas existing methods often degrade into gibberish responses.

## 2 PRELIMINARIES

### 2.1 NOTATION

Let $\theta$ denote the LLM parameters. For a pair $(x, y)$, where $x$ is the input sequence and $y = (y_1, \ldots, y_T)$ is the target sequence of length $T$, let $y_{<t} = (y_1, \ldots, y_{t-1})$ denote the prefix up to

the $t$-th token. We use $\circ$ for string concatenation. For $t \in \{1, \dots, T\}$, the model defines the next-token distribution $p(\cdot \mid x \circ y_{<t}; \theta)$ and assigns probability $p(y_t \mid x \circ y_{<t}; \theta)$ to token $y_t$. We write $\mathrm{KL}(P\|Q)$ for the Kullback-Leibler divergence from distribution $P$ to $Q$.

## 2.2 PROBLEM FORMULATION

In LLM unlearning, the goal is to remove the influence of a designated forget set $\mathcal{D}_\mathrm{F} \subseteq \mathcal{D}$ while preserving performance on the retain set $\mathcal{D}_\mathrm{R} \subseteq (\mathcal{D} \setminus \mathcal{D}_\mathrm{F})$, where $\mathcal{D}$ is the pre-training data of a pre-trained model parameterized by $\theta$. This can be formulated as optimizing a trade-off between unwanted knowledge forgetting and utility retaining:

$$\min_{\theta} \; \lambda \, \mathbb{E}_{(x,y)\sim\mathcal{D}_\mathrm{F}} \big[ \mathcal{L}_\mathrm{F}(y \mid x; \theta) \big] + \mathbb{E}_{(x,y)\sim\mathcal{D}_\mathrm{R}} \big[ \mathcal{L}_R(y \mid x; \theta) \big], \tag{1}$$

where $\mathcal{L}_\mathrm{F}$ is a forget loss encouraging removal of knowledge from $\mathcal{D}_\mathrm{F}$, $\mathcal{L}_\mathrm{R}$ is a retain loss preserving utility on $\mathcal{D}_\mathrm{R}$, and $\lambda \geq 0$ is a hyperparameter controlling the relative importance of forgetting and retaining. An effective unlearning method should suppress the model's capability on $\mathcal{D}_\mathrm{F}$ while maintaining performance on $\mathcal{D}_\mathrm{R}$, ideally matching the outcome of retraining from scratch on $\mathcal{D} \setminus \mathcal{D}_\mathrm{F}$ but at substantially lower cost.

## 2.3 BASELINE LLM UNLEARNING METHODS

We focus on parameter-optimization approaches (Yao et al., 2023; Maini et al., 2024; Zhang et al., 2024b; Liu et al., 2024b; Jia et al., 2024; Jin et al., 2024), which remain the dominant paradigm for LLM unlearning. This class of methods is particularly aligned with scenarios such as the *right to be forgotten*, *copyrighted material*, and *hazardous knowledge* removal, since they directly update a model's parameters rather than preserving its original state (Zhang et al., 2024a).

**Forget Loss.** We consider several representative baselines: Gradient Ascent (GA) (Yao et al., 2023), Negative Preference Optimization (NPO) (Zhang et al., 2024b), IDK Fine-tune (IDK) (Maini et al., 2024), Direct Preference Optimization (DPO) (Zhang et al., 2024b), and Maximizing Entropy (ME) (Yuan et al., 2024). Among these, IDK and DPO are applicable only to QA-style datasets because they require rejection templates and positive examples, respectively. More details of all baseline methods are provided in Appendix B.

**Retain Loss.** While forget losses focus on removing knowledge from the forget set, effective unlearning also requires preserving model utility. To this end, regularization on the retain set is often applied. We include two widely used retain losses below (Maini et al., 2024; Zhang et al., 2024b; Liu et al., 2024b; Jia et al., 2024); two additional variants (Yuan et al., 2024; Li et al., 2024) are provided in Appendix B:

- **Grad Descent (GD)**: standard cross-entropy loss at the output-level that performs gradient descent on the retain set, as follows:

$$\mathcal{L}_\mathrm{GD}(\mathcal{D}_\mathrm{R}; \theta) = \mathbb{E}_{(x,y)\sim\mathcal{D}_\mathrm{R}} \left[ \frac{1}{T} \sum_{t=1}^{T} -\log p(y_t | x \circ y_{<t}; \theta) \right]. \tag{2}$$

- **Kullback-Leibler Divergence (KL)**: minimizes the divergence of the prediction distribution between the unlearned model and the base model, denoted as $\theta_\mathrm{base}$ on the retain set, ensuring behavior remains consistent, as follows:

$$\mathcal{L}_\mathrm{KL}(\mathcal{D}_\mathrm{R}; \theta; \theta_\mathrm{base}) = \mathbb{E}_{(x,y)\sim\mathcal{D}_\mathrm{R}} \left[ \frac{1}{T} \sum_{t=1}^{T} \mathrm{KL}(p(\cdot | x \circ y_{<t}; \theta_\mathrm{base}) \| p(\cdot | x \circ y_{<t}; \theta)) \right]. \tag{3}$$

**Combined baselines.** By pairing forget losses with retain losses, we obtain the standard baselines used in prior work, including $\mathrm{GA_{GD}}$, $\mathrm{GA_{KL}}$, $\mathrm{NPO_{GD}}$, $\mathrm{NPO_{KL}}$, $\mathrm{DPO_{GD}}$, $\mathrm{DPO_{KL}}$, $\mathrm{IDK_{GD}}$, and $\mathrm{IDK_{KL}}$.

## 3 METHOD

Our ASU reframes unlearning as self-distillation: the goal is to suppress recall of unwanted factual information while keeping coherence and general utility intact. We construct a *forget-teacher* by

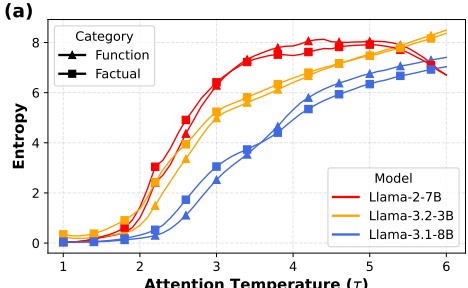 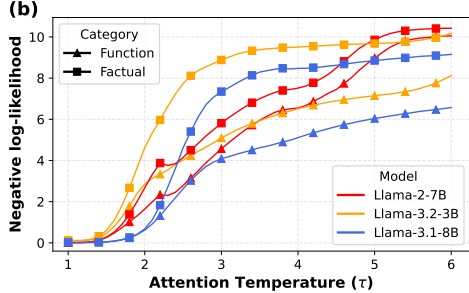

Figure 2: Effect of increasing attention temperature $\tau$. (a) Higher $\tau$ raises prediction entropy, making the model less certain about the ground-truth answer. (b) As $\tau$ grows, the average negative log-likelihood increases more sharply for *factual tokens* than for *function tokens*, indicating that recalling factual tokens depends on precise lexical attention, while function tokens are less sensitive and easier to recall.

raising the softmax temperature inside each self-attention module of the base model, which flattens attention and weakens lexical-level and semantic-level associations. This forget-teacher introduces no external models and adds no parameters beyond a single temperature, remains fixed throughout training, and is applied exclusively to the forget set. The student is trained to align with the teacher on the forget set, while a retain loss enforces preservation of the base model's utility on the retain set. We next describe the forget-teacher mechanism and the unlearning objective.

## 3.1 FORGET-TEACHER MECHANISM

In a decoder-only Transformer, each layer's multi-head self-attention (MSA) assigns weights over the prefix (earlier tokens in the input) so each token can attend to previous tokens. We form the forget-teacher by inserting a temperature $\tau \geq 1$ into the attention logits of every layer $\ell$ and head $h$. For head $h$, let $Q_h, K_h, V_h$ denote the query, key, and value matrices, and let $d_k$ be the key dimension. We define

$$\text{Attention}(Q_h, K_h, V_h; \tau) = \text{Softmax}\left(\frac{Q_h K_h^\top}{\tau\sqrt{d_k}}\right)V_h. \tag{4}$$

Setting $\tau > 1$ flattens the attention distribution by increasing entropy, thereby weakening token-to-token associations as well as their semantic representations that facilitate recall of factual information encoded in the forget set, while $\tau = 1$ recovers the base model behavior. All other components (projections, feed-forward blocks, and layer norms) remain unchanged. The forget-teacher is frozen and used solely to generate unlearning targets on the forget set.

Intuitively, increasing $\tau$ makes each attention head less selective, distributing focus more evenly across the prefix. Since base models typically exhibit low-entropy attention, smoothing weakens lexical-level and semantic-level dependencies, thereby suppressing targeted recall. As $\tau \to \infty$, the softmax approaches uniform, each head outputs the mean of past values, and the model loses the ability to precisely attend to previous relevant tokens and their representations, yielding a high-entropy distribution and incoherent outputs. This demonstrates the existence of some $\tau > 1$ that achieves the unlearning objective. We therefore treat $\tau$ as a hyperparameter that trades off forgetting efficacy against coherence: higher $\tau$ enforces stronger suppression but risks gibberish. For each task, we select a finite $\tau$ large enough to suppress factual recall on the forget set yet small enough to preserve coherence. For further details on temperature selection, refer to Appendix G.

For ASU to work, the forget-teacher should reduce the model's confidence in **factual tokens** (i.e., answer tokens that encode factual information which are *unwanted* and should be unlearned) while maintaining relatively stronger confidence in **function tokens** (i.e., grammatical tokens that ensure coherence but carry no factual information, e.g., "is," "are," "the") that support coherent language generation. In essence, smoothing ought to suppress memorized facts within the forget set while minimally disturbing core syntactic structure.

To test this, we design an experiment on the TOFU benchmark (Maini et al., 2024). Each forget instance in TOFU is a question-answer pair $(x, y)$, where we annotate the answer $y$ using GPT-4o

to distinguish factual tokens from function tokens (Zhou et al., 2025); see Appendix N for the exact instruction. We then apply attention smoothing to construct the forget-teacher, feed the concatenated sequence $x \circ y$ into it, and compute the average of negative log-likelihood and entropy for the two token types under varying temperatures.

As shown in Figure 2a, increasing $\tau$ raises entropy, indicating greater uncertainty about the ground-truth answer for both factual and function tokens, an effect we seek for unlearning. Whereas in Figure 2b, the negative log-likelihood increases far more sharply for *factual* tokens than for *function* tokens, implying that attention distribution is more essential for factual tokens compared to function tokens. Importantly, the forget-teacher assigns lower negative log-likelihood values to function tokens compared to factual ones, showing that it preserves syntax while suppressing factual recall. This explains why ASU can preserve utility and produce coherent outputs, in contrast to baselines that often collapse into gibberish.

## 3.2 UNLEARNING OBJECTIVE

Attention smoothing weakens lexical-level and semantic-level associations, so it should be applied exclusively to the forget set that encodes unwanted factual knowledge; applying it more broadly risks degrading useful associations needed for general tasks. In practice, we only distill knowledge from the forget-teacher on the forget set. For the forget set $\mathcal{D}_F$, we minimize the KL divergence between the outputs of $\theta$ and those of the attention-smoothed model $\theta_\tau$, where $\tau$ is the temperature applied to the attention softmax. This objective guides the model to reproduce the smoothed, association-suppressed behavior on forget-set inputs. We define the forget loss as follows:

$$\mathcal{L}_{\text{ASU}}(\mathcal{D}_F; \theta; \theta_\tau) = \mathbb{E}_{(x,y)\sim\mathcal{D}_F}\left[\frac{1}{T}\sum_{t=1}^{T}\text{KL}\Big(p(\cdot \mid x \circ y_{<t}; \theta_\tau)\|p(\cdot \mid x \circ y_{<t}; \theta)\Big)\right]. \tag{5}$$

Finally, we apply GD-based 2 or KL-based 3 regularization on the retain set, yielding $\text{ASU}_{\text{GD}}$ and $\text{ASU}_{\text{KL}}$ approaches. Our representation steering approach is described in Appendix F.

## 4 EXPERIMENTS

We evaluate three scenarios across standard datasets: (i) Right to Be Forgotten with TOFU, including continual and real-world variants; (ii) copyrighted-content removal with MUSE; and (iii) hazardous-knowledge unlearning with WMDP, whose results are provided in the Appendix F. We describe each setup in the following sections. The selected temperatures are detailed in Appendix H.

### 4.1 RIGHT TO BE FORGOTTEN UNLEARNING SCENARIO

#### 4.1.1 FICTITIOUS UNLEARNING SCENARIO

**Setup.** TOFU (Maini et al., 2024) is a controlled benchmark for *sample-level* unlearning in LLMs. It constructs a synthetic corpus of *200* fictitious authors, each with *20* question–answer pairs. A target model (e.g., Llama-2-Chat-7B) is fine-tuned on the full corpus to induce memorization; unlearning then removes a designated subset while preserving utility on related content. The benchmark defines three tasks, forget01, forget05, and forget10, which require forgetting {1%, 5%, 10%} of authors (2/10/20 authors), respectively; the complement serves as the *retain* set. Two auxiliary sets, *Real Authors* and *World Facts*, are also provided to evaluate general knowledge.

**Evaluation Metrics.** Following previous works (Yuan et al., 2024; Maini et al., 2024), we use ROUGE-L recall (R), Probability (P), Truth Ratio (TR), Cosine Similarity (CS), Entailment Score (ES), and Token Entropy (TE). **Model Utility (MU)** is the harmonic mean of $\{R, P, \max(0, 1 - \text{TR}), \text{CS}, \text{ES}, \text{TE}\}$ on the retain set and the *Real Authors* and *World Facts* sets. **Forget Efficacy (FE)** is the harmonic mean of $\{1 - R, 1 - P, 1 - \min(\text{TR}, 1/\text{TR}), 1 - \text{ES}, \text{TE}\}$ on the forget set. Higher MU/FE indicate better utility/forgetting. See Appendix C.1 for details.

**Performance on TOFU.** Table 1 summarizes results across the three TOFU unlearning tasks. Our ASU variants (i.e., $\text{ASU}_{\text{GD}}$, and $\text{ASU}_{\text{KL}}$) consistently deliver the best overall performance, as reflected by their dominance in bold and underlined scores across both FE and MU. While $\text{IDK}_{\text{AP}}$

Table 1: Results of unlearning methods on the TOFU benchmark. *Higher is better for all metrics.* We report Model Utility (MU), Forget Efficacy (FE), and their **Average (Avg.)** across the three TOFU tasks. Best scores are in **bold**, and second-best are underlined. All results are reported in percentages. We show the detailed results for each metric on the retain set and the forget set for three tasks in the Appendix Table 11 and Table 12.

| Method | forget01 | | | forget05 | | | forget10 | | |
|---|---|---|---|---|---|---|---|---|---|
| | MU | FE | Avg. | MU | FE | Avg. | MU | FE | Avg. |
| Base | 75.81 | 3.09 | 39.45 | 75.85 | 3.19 | 39.52 | 75.85 | 3.19 | 39.52 |
| **Divergence-based** | | | | | | | | | |
| $GA_{GD}$ | 66.59 | 69.46 | 68.02 | 29.25 | 3.89 | 16.57 | 50.29 | 0.01 | 25.15 |
| $GA_{KL}$ | 67.83 | 68.73 | 68.28 | 20.13 | 5.39 | 12.76 | 54.38 | 11.17 | 32.78 |
| $NPO_{GD}$ | 64.10 | 71.14 | 67.62 | 56.62 | 73.31 | 64.97 | 56.58 | 73.04 | 64.81 |
| $NPO_{KL}$ | 64.19 | 70.71 | 67.45 | 57.70 | 73.35 | 65.52 | 57.00 | 70.37 | 63.68 |
| **Convergence-based** | | | | | | | | | |
| $DPO_{GD}$ | 75.68 | 42.91 | 59.29 | 0.00 | 77.15 | 38.58 | 0.00 | 74.31 | 37.15 |
| $DPO_{KL}$ | 75.63 | 42.70 | 59.16 | 0.00 | 77.22 | 38.61 | 0.00 | 74.44 | 37.22 |
| $IDK_{AP}$ | 75.69 | 60.29 | 67.99 | **75.23** | 60.88 | 68.05 | **74.24** | 61.27 | 67.76 |
| $IDK_{GD}$ | 66.94 | 61.03 | 63.99 | 0.00 | 70.18 | 35.09 | 5.26 | 58.80 | 32.03 |
| $IDK_{KL}$ | 67.14 | 61.16 | 64.15 | 0.00 | 70.18 | 35.09 | 7.52 | 59.06 | 33.29 |
| $ME_{GD}$ | 72.48 | 75.04 | 73.76 | 74.96 | 70.15 | 72.56 | 73.36 | 45.95 | 59.65 |
| $ME_{KL}$ | 73.82 | 67.04 | 70.43 | 74.43 | 70.44 | 72.43 | 73.84 | 44.29 | 59.06 |
| $ASU_{GD}$ | 76.79 | 82.20 | 79.50 | 73.62 | 77.58 | 75.60 | 73.82 | **78.72** | **76.27** |
| $ASU_{KL}$ | **77.13** | **83.08** | **80.10** | 74.18 | **77.84** | **76.01** | 73.27 | 78.16 | 75.71 |

attains slightly higher MU on forget05 (75.23) and forget10 (74.24), ASU achieves comparable utility (e.g., $ASU_{KL}$ reaches 74.18 and 73.27, respectively) while substantially outperforming $IDK_{AP}$ on forgetting. Specifically, $ASU_{KL}$ attains FE of 77.84 on forget05 and 78.16 on forget10, compared to 60.88 and 61.27 for $IDK_{AP}$, a nearly 30% increase of FE (60.88 → 77.84 and 61.27 → 78.16). These results highlight ASU's ability to maintain strong utility while achieving state-of-the-art FE, offering the most effective and stable trade-off among all methods.

### 4.1.2 CONTINUAL UNLEARNING SCENARIO

**Setup.** We study a continual unlearning setup where a base model is subjected to a sequence of unlearning requests, each removing a disjoint subset of authors in the TOFU benchmark while preserving utility on the remaining *retain* data (Yuan et al., 2024). Unlike single-shot evaluations, this setting mirrors rolling Right-to-be-Forgotten requests in practice and exposes cumulative degradation effects as utility preservation becomes progressively harder with each step, due to a shrinking retain pool and shifting distributional coverage. Concretely, we run sequences where each step removes either `forget01` (1%), `forget05` (5%), or `forget10` (10%) of the authors, For `forget01` and `forget05` we run 10 steps, resulting in cumulative removals of 10% and 50%, respectively. For `forget10` we run 9 steps, removing up to 90% of authors in total. After each step, we evaluate using the same metrics as in the TOFU task (R, P, TR, CS, ES, TE), reporting the average of **MU** on retain/general-knowledge sets and **FE** on the current forget set. For fair comparison, we chose GD as the retain loss for all of the baselines.

**Performance.** Figure 3 reports the average scores of MU and EF in continual unlearning on TOFU, where disjoint subsets of authors are removed across multiple steps. As expected, maintaining high average performance becomes increasingly difficult as the retain pool shrinks and distributional coverage narrows. GA collapses immediately across all three settings, yielding near-zero averages. In the more challenging scenarios (i.e., continual forget05 and forget10), NPO (Zhang et al., 2024b) and IDK (Maini et al., 2024) begin with moderately strong average scores, but significantly degrade with successive unlearning steps, highlighting their instability in long-horizon unlearning. DPO (Zhang et al., 2024b) and ME (Yuan et al., 2024) show more stable curves in continual unlearning steps, but start with considerably lower averages than ASU. For example, on `forget10`, ME attains scores of roughly 70 and DPO around 45, both substantially lower than ASU, which consistently maintains an average close to 75.

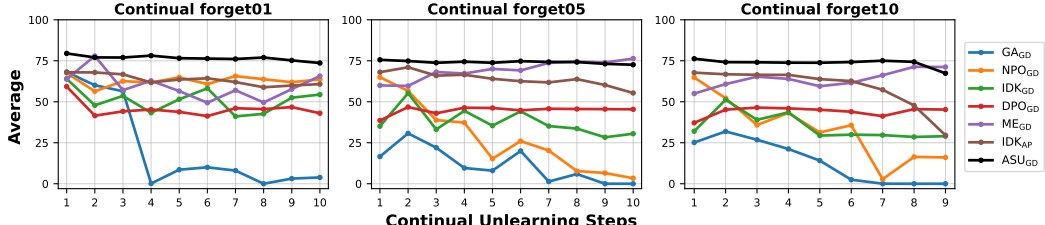

Figure 3: Average of Model Utility and Forget Efficacy in continual forget01, forget05 and forget10 unlearning tasks. We show the results for MU and FE in the Appendix Figure 5 and Figure 6.

Table 2: **Results of real-world unlearning scenario.** *Higher is better for all metrics.* Base represents the model before unlearning. Model Utility (MU) and Forget Efficacy (FE) are calculated on the neighbor set and forget set, respectively. Please see the detailed results in the Appendix Table 6.

| Method | Unlearning Task | | Downstream Tasks | | | | |
|---|---|---|---|---|---|---|---|
| | Model Utility | Forget Efficacy | ARC-c | MMLU | TruthfulQA | GSM8K | Avg. |
| Base | 61.38 | 36.83 | 56.57 | 63.84 | 36.11 | 75.51 | 58.01 |
| **Divergence-based Unlearning** | | | | | | | |
| $GA_{GD}$ | 21.76 | 65.73 | 51.37 | 58.80 | 39.29 | 27.14 | 44.15 |
| $GA_{KL}$ | 43.72 | 0.00 | 46.84 | 58.39 | 25.46 | 24.03 | 38.68 |
| $NPO_{GD}$ | 21.38 | 71.44 | 38.40 | 53.49 | 34.15 | 69.29 | 48.83 |
| $NPO_{KL}$ | 27.32 | 72.11 | 37.80 | 51.80 | 33.66 | 67.10 | 47.59 |
| **Convergence-based Unlearning** | | | | | | | |
| $DPO_{GD}$ | 0.00 | 82.45 | 50.94 | 62.16 | 31.82 | 72.48 | 54.35 |
| $DPO_{KL}$ | 3.28 | 83.48 | 50.68 | 62.00 | 31.46 | 72.18 | 54.08 |
| $IDK_{GD}$ | 0.00 | 78.40 | 52.47 | 62.48 | 32.44 | 74.53 | 55.48 |
| $ME_{GD}$ | 47.96 | 48.10 | 52.99 | 62.48 | 31.21 | 69.52 | 54.05 |
| $IDK_{AP}$ | 52.76 | 78.04 | 53.41 | 62.04 | 27.05 | 73.24 | 53.94 |
| $ASU_{GD}$ | 54.10 | 76.97 | 49.32 | 63.42 | 28.27 | 63.91 | 51.23 |
| $ASU_{KL}$ | **55.76** | **79.60** | 51.19 | 62.90 | 33.90 | 68.84 | 54.21 |

Compared to all competing methods, ASU consistently achieves the best trade-off between forget efficacy and utility preservation over long sequences of unlearning requests. Even under extreme conditions where up to 90% of authors are unlearned (`forget10`), ASU exhibits a markedly slower degradation, maintaining strong performance when other methods collapse. This robustness to continual unlearning pressure highlights ASU's suitability for real-world applications such as continual Right-to-be-Forgotten requests.

### 4.1.3 REAL-WORLD UNLEARNING SCENARIO

**Setup.** Following (Liu et al., 2025b; Yuan et al., 2024), we evaluate unlearning when the target model's training data are unknown and the knowledge to be removed is intrinsically memorized. We construct a *real-world forget set* by selecting a small cohort of real individuals with strong memorization in the target model and collecting the model's own answers to curated prompts. A disjoint cohort of comparable individuals forms the *neighbor/retain* pool; a subset is used for regularization during unlearning and the remainder for utility evaluation. To assess general utility preservation, we also report performance on standard downstream benchmarks (e.g., MMLU, ARC-c, GSM8K, TruthfulQA). We use the same metrics as in the TOFU task (R, P, TR, CS, ES, TE) and report MU on retain/general-knowledge evaluations and FE on the real-world forget set.

**Performance.** Table 2 reports results for the real-world unlearning scenario. Divergence-based methods (e.g., GA, NPO) achieve competitive forget efficacy but suffer from severe utility collapse, with most MU scores dropping to 21–28, far below the benchmark of 61.38. Convergence-based approaches (i.e., DPO, IDK) push FE even higher (up to 83.48) but collapse MU to nearly zero. *In contrast, our $ASU_{KL}$ achieves the best overall trade-off, with MU = 55.76 and FE = 79.60, outperforming all baselines on both dimensions.* $ASU_{GD}$ achieves similar results (FE = 76.97 and MU = 54.10), underscoring the robustness of ASU across retain-loss variants. Moreover, both ASU

Table 3: Performance of various unlearning methods on MUSE, considering two unlearning settings: LLaMA2-7B on News and ICLM-7B on Books.

| Method | News | | | | Books | | | |
|---|---|---|---|---|---|---|---|---|
| | Forget Efficacy | | | Model Utility | Forget Efficacy | | | Model Utility |
| | VerbMem $\mathcal{D}_F(\downarrow)$ | KnowMem $\mathcal{D}_F(\downarrow)$ | PrivLeak $(\to 0)$ | KnowMem $\mathcal{D}_R(\uparrow)$ | VerbMem $\mathcal{D}_F(\downarrow)$ | KnowMem $\mathcal{D}_F(\downarrow)$ | PrivLeak $(\to 0)$ | KnowMem $\mathcal{D}_R(\uparrow)$ |
| Base | 57.9 | 64.4 | -99.8 | 55.5 | 99.7 | 47.1 | -57.3 | 69.1 |
| Retrain | 20.2 | 32.8 | 0.0 | 56.0 | 14.4 | 30.3 | 0.0 | 68.7 |
| GA$_{GD}$ | 3.6 | 1.9 | 9.4 | 0.7 | 0.0 | 0.0 | -23.8 | 0.0 |
| GA$_{KL}$ | 6.8 | 1.0 | 43.9 | 0.0 | 0.0 | 0.0 | -24.9 | 0.0 |
| NPO$_{GD}$ | 33.7 | 54.3 | -86.0 | 50.5 | 53.2 | 36.6 | -53.8 | 61.4 |
| NPO$_{KL}$ | 33.0 | 56.2 | -85.7 | 49.3 | 54.4 | 36.7 | -54.6 | 61.4 |
| SimNPO$_{GD}$ | 41.7 | 60.0 | -99.9 | 42.8 | 25.8 | 36.7 | -54.4 | 51.6 |
| SimNPO$_{KL}$ | 43.8 | 60.7 | -99.8 | 52.0 | 13.1 | 46.9 | -41.7 | 68.1 |
| ASU$_{GD}$ | 8.3 | 48.0 | 22.8 | 46.2 | 4.9 | 19.0 | -52.3 | 58.9 |
| ASU$_{KL}$ | 8.8 | 46.8 | 59.6 | 52.2 | 5.3 | 28.6 | -51.0 | 62.5 |

variants sustain accuracy on downstream benchmarks at levels comparable to or exceeding other baselines, demonstrating that ASU effectively removes memorized real-world knowledge while preserving general utility.

## 4.2 COPYRIGHT UNLEARNING SCENARIO

**Setup.** We use MUSE (Shi et al., 2024) to assess unlearning of copyrighted content. MUSE provides two corpora (News, Books), each partitioned into three disjoint splits: forget, retain, and holdout (non-members). Each corpus includes a Verbatim set (passages) and a Knowledge set (QA derived from those passages). Following (Shi et al., 2024), the target model is fine-tuned on the union of forget and retain, and the retrain baseline is fine-tuned on retain only.

**Metrics.** Following previous works (Shi et al., 2024), we evaluate using three standard unlearning metrics: **VerbMem** (verbatim recall), **KnowMem** on both forget and retain splits (factual association and utility), and **PrivLeak** (membership leakage). Full definitions and implementation details are provided in Appendix C.2.

**Performance on MUSE.** Table 3 reports results on the MUSE benchmark under the News and Books settings. On News, GA variants (i.e, GA$_{GD}$, and GA$_{KL}$) suffer from complete utility collapse, with their KnownMem score on the retain set dropping close to zero. Therefore, their forgetting efficacy is less meaningful to interpret. Considering the remaining baselines (NPO and SimNPO variants), *ASU variants provide the best overall trade-off between FE and MU*. In particular, ASU$_{GD}$ achieves the strongest FE performance, while ASU$_{KL}$ delivers comparable FE to ASU$_{GD}$ but clearly surpasses all baselines and preserves the highest MU, attaining a KnowMem score of 52.2 on $\mathcal{D}_R$.

On the Books setting, GA variants once again collapse in utility, with KnowMem $\mathcal{D}_R$ dropping to zero. NPO and SimNPO variants achieve only partial forgetting, either leaving VerbMem high (e.g., NPO$_{KL}$ = 54.4) or retaining substantial KnowMem (e.g., SimNPO$_{KL}$ = 46.9), indicating incomplete unlearning. *In contrast, our ASU variants achieve a more favorable trade-off between FE and MU*. ASU$_{GD}$ provides the strongest forgetting across all metrics, while ASU$_{KL}$ provides the best overall balance, delivering effective forgetting (VerbMem = 5.3, KnowMem = 28.6, PrivLeak = -51.0) while maintaining the comparable utility (KnowMem = 62.5). These results demonstrate that ASU generalizes effectively across different domains, preserving utility while ensuring stronger forgetting than existing baselines.

## 5 ABLATION STUDIES

### 5.1 IMPACT OF SMOOTHING PARTIAL LAYERS ON FACTUAL VS. FUNCTION TOKENS

We previously showed in Section 3.1 that smoothing attention across all layers reduces the model's NLL in factual tokens. A plausible reason is that LLMs encode syntactic operations (function tokens) and factual knowledge in fundamentally different ways. Functional tokens support grammatical structure and appear extremely frequently during pre-training, which makes their embeddings stable and resistant to perturbations in shallow-layer attention. In contrast, factual knowledge ap-

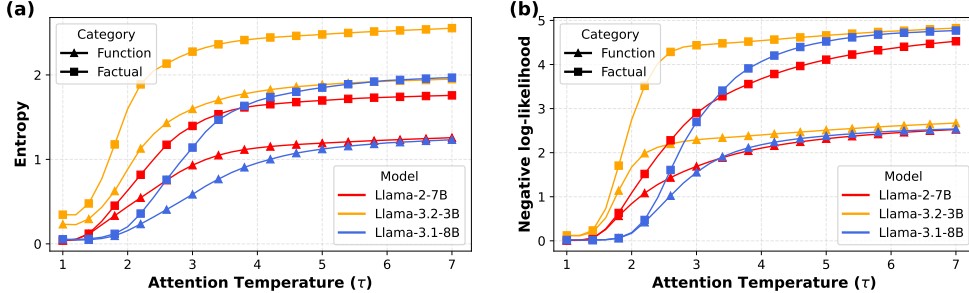

Figure 4: Effect of increasing attention temperature $\tau$ for consecutive shallow layers.

pears only in a small portion of the corpus and relies on precise lexical and semantic associations. These associations are considerably more fragile: smoothing the attention of shallow layers is sufficient to disrupt the recall of factual tokens while leaving the syntactic scaffold largely unaffected.

To validate this explanation, we conducted an additional experiment where we smooth only the shallow layers (e.g., layers 6–8). We focus on shallow layers because prior work shows that earlier transformer layers play a more important role in factual associations in LLMs (Meng et al., 2022; Guo et al., 2025). Under this setting, both entropy and NLL for factual tokens increase much more sharply than for functional tokens, as shown in Figure 4. This result confirms that factual tokens depend more heavily on precise attention patterns. Please refer to the Appendix I for a comprehensive set of ablations examining how smoothing different subsets of layers affects factual and functional token behavior. When we use this shallow-smoothed model as the forget-teacher, we obtain nearly the same forget efficacy and model utility on TOFU tasks (Table 4) as our default setting that smooths all layers (Table 1).

Table 4: ASU results on TOFU with smoothing applied only to layers $6, 7, 8$.

| Task | Method | MU | FE | Avg. |
|---|---|---|---|---|
| **forget01** | $\text{ASU}_{\text{GD}}$ | 75.74 | 79.52 | 77.63 |
| | $\text{ASU}_{\text{KL}}$ | 75.77 | 80.45 | 78.11 |
| **forget05** | $\text{ASU}_{\text{GD}}$ | 71.82 | 77.62 | 74.72 |
| | $\text{ASU}_{\text{KL}}$ | 72.39 | 77.49 | 74.94 |
| **forget10** | $\text{ASU}_{\text{GD}}$ | 71.64 | 77.14 | 74.39 |
| | $\text{ASU}_{\text{KL}}$ | 70.89 | 76.90 | 73.90 |

Table 5: Performance of ASU integrated with $\text{IDK}_{AP}$ on TOFU.

| Task | Method | MU | FE | Avg. |
|---|---|---|---|---|
| **forget01** | $\text{ASU}_{\text{GD}}$ | 76.67 | 80.69 | 78.68 |
| | $\text{ASU}_{\text{KL}}$ | 76.75 | 80.72 | 78.74 |
| **forget05** | $\text{ASU}_{\text{GD}}$ | 76.15 | 83.50 | 79.82 |
| | $\text{ASU}_{\text{KL}}$ | 76.24 | 83.28 | 79.76 |
| **forget10** | $\text{ASU}_{\text{GD}}$ | 75.60 | 86.94 | 81.27 |
| | $\text{ASU}_{\text{KL}}$ | 75.61 | 86.77 | 81.19 |

## 5.2 INTEGRATING ASU WITH REFUSAL-STYLE OUTPUTS

Since the refusal-style output can only be applied to QA datasets (e.g., TOFU) and can not be used in non-QA datasets (e.g., MUSE and WMDP), we follow prior work (GA, NPO, ME) and do not train ASU itself to refuse. To further demonstrate the flexibility and effectiveness of our method, we combine ASU with a refusal-based baseline, $\text{IDK}_{AP}$, and train the model to generate refusal-style outputs on the TOFU benchmark (using the same setup as Table 1). Table 5 shows that this combined approach yields consistently higher MU and FE scores than the original baselines in Table 1. For instance, on the most challenging task, forget10, both $\text{ASU}_{\text{GD}}$ and $\text{ASU}_{\text{KL}}$ achieve MU above 75 and FE above 80, whereas $\text{IDK}_{AP}$ alone reaches only MU 74.24 and FE 61.27. This indicates that ASU effectively removes factual knowledge that $\text{IDK}_{AP}$ alone fails to erase, while preserving the model's ability to produce refusal-style responses on the forget set.

## 5.3 STABILITY OF ASU UNDER VARIOUS TEMPERATURE VALUES

To further assess the stability of ASU with respect to the attention temperature, we conduct additional experiments on the TOFU `forget05` task using a range of temperature values $\tau \in \{2.0, 2.2, 2.4, 2.6, 2.8, 3.0\}$ (our main results in Table 1 use $\tau = 2.3$). The full results are reported in Table 9 in Appendix. As shown in Table 9, ASU remains stable across a broad interval: temperatures

between 2.0 and 2.8 yield highly consistent MU and FE for both $\text{ASU}_{\text{GD}}$ and $\text{ASU}_{\text{KL}}$. These results demonstrate that ASU is robust to the choice of temperature within a wide and practical range.

## 6 RELATED WORK

**Machine Unlearning.** Machine Unlearning (MU) seeks to remove the effect of specific data or facts without full retraining, which is often prohibitively expensive (Cao & Yang, 2015; Bourtoule et al., 2021; Ginart et al., 2019; Golatkar et al., 2020). Existing works provide approximate unlearning methods (Warnecke et al., 2021; Izzo et al., 2021; Sekhari et al., 2021), influence-function approaches (Koh & Liang, 2017), and second-order optimization (Jia et al., 2024). MU has been studied across diverse domains such as image classification (Neel et al., 2021), text-to-image generation (Gandikota et al., 2023; Kumari et al., 2023), federated settings (Wang et al., 2022; Halimi et al., 2022), and graph neural networks (Chen et al., 2022; Wu et al., 2023), and is especially relevant for LLMs where retraining a model from scratch is infeasible.

**LLM unlearning.** Motivated by privacy regulations (Regulation, 2016; Pardau, 2018) such as the "right to be forgotten" (Rosen, 2011; Dang, 2021), LLM unlearning has become an active research area. The main approaches fine-tune the model in a forgotten set to obtain an unlearned version including gradient-ascent based methods (Jang et al., 2022; Yao et al., 2024b; Tunstall et al., 2023; Ishibashi & Shimodaira, 2023; Fan et al., 2024; Maini et al., 2024; Tamirisa et al., 2024; Zhou et al., 2025), preference optimization methods (Zhang et al., 2024b; Mekala et al., 2024; Wang et al., 2024a; 2025a), knowledge distillation (Dong et al., 2024; Lu et al., 2024; Yao et al., 2024a; Jia et al., 2024; Tian et al., 2024; Gu et al., 2024; Eldan & Russinovich, 2023a), influence functions (Jia et al., 2023; Grosse et al., 2023; Zhao et al., 2024; Liu et al., 2024b; Dang et al., 2025; Wang et al., 2024b; 2025b; Sakarvadia et al., 2025), activation steering (Li et al., 2024; Dang et al., 2025), localized edits (Guo et al., 2025; Wuerkaixi et al., 2025; Fan et al., 2025; Wang et al., 2025c; Gao et al., 2025; Ding et al., 2025). Other works focus on inference-time unlearning, including contrastive decoding (Huang et al., 2024a; Ji et al., 2024), in-context unlearning (Pawelczyk et al., 2023; Muresanu et al., 2024), guardrails (Thaker et al., 2024; Bhaila et al., 2024), task vector–based methods (Ilharco et al., 2022; Liu et al., 2024c; Dou et al., 2024), and input pre-processing (Gao et al., 2024; Liu et al., 2024a). However, most of these methods do not modify the LLM parameters, so the resulting system cannot be released as an open model and may still raise security concerns in black-box settings (Shi et al., 2023; Zade et al., 2025). In this work, we investigate the role of attention in unlearning from a novel perspective.

**Adjusting Attention.** Beyond unlearning, attention adjustments, through temperature scaling or normalization, have been applied across diverse tasks, such as improving translation (Araabi et al., 2024; Henry et al., 2020), accelerating sequence labeling (Dufter et al., 2020), smoothing teacher signals for summarization distillation (Zhang et al., 2022), improving stability by avoiding entropy collapse (Zhai et al., 2023), maintaining selective focus in long-context reasoning (Veličković et al., 2024), tuning sparsity per query in LLMs (Zhang et al., 2024c), and aiding cross-domain few-shot transfer in vision (Zou et al., 2024). Moreover, previous work shows that the ablation of several attention heads can impact safety (Zhou et al., 2024b). To the best of our knowledge, its effect on unlearning has not yet been explored.

## 7 CONCLUSION

We introduced ASU, a method that reframes unlearning as self-distillation from a forget-teacher constructed by raising the softmax temperature in attention. By flattening attention and weakening the lexical-level and semantic-level associations that drive factual recall, ASU effectively erases memorized content while keeping responses on forget prompts coherent. Extensive experiments across various scenarios show that ASU reaches strong forget efficacy with minimal utility loss, and unlike prior divergence-based or convergence-based methods, it avoids gibberish outputs or under-forgetting. These findings position ASU method as a simple, practical path for unlearning in LLMs and for safer model release.

ETHICS STATEMENT

This work investigates unlearning techniques for LLMs, with the goal of enabling models to forget specific undesirable or sensitive knowledge while retaining general utility. Our experiments are conducted on publicly available datasets and do not involve private or personally identifiable information. We recognize that unlearning methods may raise ethical concerns if misused, for example by selectively erasing knowledge in ways that distort truth, suppress marginalized perspectives, or enable malicious applications. To mitigate these risks, we focus on controlled benchmarks, transparently report our methodology and limitations, and emphasize that unlearning should be applied responsibly, in alignment with broader principles of trustworthy and safe AI.

REPRODUCIBILITY STATEMENT

We have taken several steps to facilitate the reproducibility of our results. All datasets used in our experiments are publicly available. We provide detailed descriptions of baselines and evaluation protocols in the main text and appendix. Our code, including scripts to reproduce the experiments and generate the reported figures and tables, is available at Github.

ACKNOWLEDGMENTS

This paper was supported by the U.S. National Science Foundation (NSF) under Award Numbers IIS-2211897, IIS-2504264, and IIS-2504263. Any opinions, findings, and conclusions or recommendations expressed in this material are those of the authors and do not necessarily reflect the views of the National Science Foundation.

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

APPENDIX

# A  PROOF

## A.1  NOTATIONS

Let $V$ be a finite vocabulary, and let $\theta$ denote the parameters of a pretrained decoder-only language model. For any input–output pair $(x, y)$, where

$$x = (x_1, \ldots, x_L) \quad \text{and} \quad y = (y_1, \ldots, y_T),$$

the model defines conditional probabilities

$$p_\theta(y_t \mid x \circ y_{<t}), \qquad t = 1, \ldots, T.$$

For any $(x, y) \in D_F$ (*forget set*), we partition the target positions into

$$F \subseteq \{1, \ldots, T\} \quad \text{(factual positions)}, \qquad G = \{1, \ldots, T\} \setminus F \quad \text{(function positions)}.$$

Factual positions correspond to the tokens that encode the unwanted information to be removed, whereas function positions refer to tokens that serve primarily syntactic or structural roles within the sequence.

## A.2  SELF-ATTENTION AND TEMPERATURE

Consider a single Transformer layer with one attention head (layer and head indices are omitted for clarity; the argument applies to each head independently). For a position $t$, let $q_t \in \mathbb{R}^d$ denote the query vector, and let $k_i, v_i \in \mathbb{R}^d$ be the key and value vectors for all positions $i \leq t$.

The attention logits are

$$a_{t,i} := \frac{\langle q_t, k_i \rangle}{\sqrt{d}}, \qquad i = 1, \ldots, t,$$

and the standard attention weights (with temperature set to 1) are

$$\alpha_{t,i} = \frac{\exp(a_{t,i})}{\sum_{j=1}^{t} \exp(a_{t,j})}.$$

The corresponding attention output is

$$z_t = \sum_{i=1}^{t} \alpha_{t,i} v_i.$$

We introduce a temperature parameter $\tau \geq 1$ and define the smoothed attention weights

$$\alpha_{t,i}(\tau) = \frac{\exp(a_{t,i}/\tau)}{\sum_{j=1}^{t} \exp(a_{t,j}/\tau)},$$

with attention output

$$z_t(\tau) = \sum_{i=1}^{t} \alpha_{t,i}(\tau) v_i.$$

When $\tau = 1$, the model recovers the base attention: $\alpha_{t,i}(1) = \alpha_{t,i}$ and $z_t(1) = z_t$. For $\tau > 1$, the distribution $\alpha_t(\tau)$ becomes strictly flatter than $\alpha_t$ due to the scaling of all logit differences by $1/\tau$.

We define the *attention-smoothed teacher model* $\theta_\tau$ as the model obtained by applying temperature $\tau$ in all attention heads while keeping all other components (feed-forward layers, layer norms, and output projection) unchanged.

## A.3 OUTPUT LAYER AND TOKEN PROBABILITIES

Let $W \in \mathbb{R}^{|V| \times d}$ and $b \in \mathbb{R}^{|V|}$ denote the output projection matrix and bias. At position $t$, let $h_t$ be the hidden representation produced by the Transformer (which incorporates the attention output $z_t$ through the subsequent layers).

For each token $w \in V$, the model computes the logit

$$\ell_t(w; \theta) = \langle W_w, h_t \rangle + b_w,$$

and the corresponding conditional probability

$$p_\theta(w \mid x \circ y_{<t}) = \frac{\exp(\ell_t(w; \theta))}{\sum_{u \in V} \exp(\ell_t(u; \theta))}.$$

For the attention-smoothed teacher model $\theta_\tau$, applying temperature $\tau$ only inside the attention mechanism yields modified hidden states $h_t(\tau)$, which produce logits

$$\ell_t(w; \theta_\tau)$$

and token probabilities

$$p_{\theta_\tau}(w \mid x \circ y_{<t}).$$

## A.4 NOTIONS OF "FORGETTING" AND "FLUENCY"

We define two properties of interest: the removal of specific factual content and the preservation of normal language behavior.

### A.4.1 FORGETTING

Fix a forget example $(x, y) \in D_F$ and a factual position $t \in F$. Let $y_t^\star$ denote the factual token to be removed (for example, the correct entity name in TOFU).

We say that the smoothed model $\theta_\tau$ forgets this fact at position $t$ if

$$p_{\theta_\tau}(y_t^\star \mid x \circ y_{<t}) \leq \epsilon_F,$$

for some small threshold $\epsilon_F > 0$ (roughly the level of random guess accuracy among plausible entities).

At the sequence level, forgetting holds on $D_F$ when the average

$$- \log p_{\theta_\tau}(y_t^\star \mid x \circ y_{<t})$$

over all $(x, y) \in D_F$ and all $t \in F$ is at least a target value $L_F$, meaning the model assigns low probability to the factual tokens.

### A.4.2 FLUENCY

For function positions $t \in G$, we require that the model continue to assign high probability to the correct function tokens, which reflect grammar and structure.

We say that $\theta_\tau$ preserves fluency on $(x, y)$ if

$$- \log p_{\theta_\tau}(y_t \mid x \circ y_{<t}) \leq - \log p_\theta(y_t \mid x \circ y_{<t}) + \delta_G, \qquad t \in G,$$

for a tolerance $\delta_G > 0$.

At the sequence level, fluency is preserved if the average cross-entropy on function tokens increases by at most $\delta_G$.

## A.5 ASSUMPTIONS

To show that attention smoothing can remove specific facts while keeping normal language behavior, we introduce structural assumptions on how factual and function tokens depend on attention.

### A.5.1    ASSUMPTION A1 (FACTUAL TOKENS REQUIRE PRECISE ATTENTION)

For every factual position $t \in F$, there exists a subset $S_t \subseteq \{1, \dots, t\}$ with $|S_t| \ll t$ such that the base attention places most of its mass on $S_t$:

$$\sum_{i \in S_t} \alpha_{t,i} \geq \gamma, \qquad \gamma \in (0, 1).$$

The value vectors at positions in $S_t$ contain the main signal supporting the factual token $y_t^\star$, while positions outside $S_t$ contribute less to this signal.

Define

$$z_t = \sum_{i=1}^{t} \alpha_{t,i} v_i, \qquad z_t(S_t) := \frac{1}{|S_t|} \sum_{i \in S_t} v_i, \qquad z_t(\bar{S}_t) := \frac{1}{t - |S_t|} \sum_{i \notin S_t} v_i,$$

and

$$z_t^{\mathrm{unif}} := \frac{1}{t} \sum_{i=1}^{t} v_i = \frac{|S_t|}{t} z_t(S_t) + \frac{t - |S_t|}{t} z_t(\bar{S}_t).$$

We assume that concentrating attention on $S_t$ increases the factual logit margin, and that the uniform attention mixture reduces this margin by at least $m_F > 0$:

$$\langle W_{y_t^\star} - W_u, \; z_t - z_t^{\mathrm{unif}} \rangle \geq m_F \qquad \text{for all } u \neq y_t^\star.$$

Thus, moving the attention distribution away from the sharp pattern concentrated on $S_t$ toward a flatter (uniform) distribution decreases the logit margin of the factual token $y_t^\star$.

### A.5.2    ASSUMPTION A2 (FUNCTION TOKENS ARE LESS ATTENTION-SENSITIVE)

For each function position $t \in G$, we assume that the correct token $y_t$ depends on a broad mixture of value vectors rather than on a small set of positions. In other words, predicting $y_t$ does not rely on a sharp attention pattern.

Formally, let $\ell_t(y_t; \theta; z)$ denote the logit of $y_t$ when the attention output at position $t$ is $z$. Assume the logit is smooth with respect to $z$ and satisfies an $L$-Lipschitz bound:

$$|\ell_t(y_t; \theta; z) - \ell_t(y_t; \theta; z')| \leq L \, \|z - z'\|_2 \qquad \text{for all } z, z'.$$

We also assume that the convex combinations of $\{v_i\}_{i \leq t}$ do not have strong changes in the direction of $W_{y_t}$. Thus, shifting the attention weights from a sharper pattern toward a smoother one (such as closer to uniform) causes only a small change in $\ell_t(y_t)$.

### A.5.3    ASSUMPTION A3 (NON-DEGENERATE LOGITS FOR FACTUAL TOKENS)

For each factual position $t \in F$, the base model assigns a clear margin to the correct factual token $y_t^\star$. Formally,

$$\ell_t(y_t^\star; \theta) - \max_{u \neq y_t^\star} \ell_t(u; \theta) \geq \Delta_F > 0.$$

This ensures that factual recall in the base model is supported by a positive logit gap.

### A.5.4    ASSUMPTION A4 (CONTINUITY)

For every position $t$, the attention-smoothed hidden state $h_t(\tau)$ and the logits $\ell_t(w; \theta_\tau)$ vary continuously with respect to the temperature parameter $\tau$.

This holds for standard Transformer layers, since attention, linear transformations, and activation functions are continuous.

## A.6 LEMMAS

### A.6.1 LEMMA 1 (DIRECTION OF ATTENTION CHANGES UNDER TEMPERATURE)

Fix a position $t$ and attention logits $a_{t,1}, \ldots, a_{t,t} \in \mathbb{R}$. For $\tau > 0$, define the temperature-scaled attention weights

$$\alpha_{t,i}(\tau) = \frac{\exp(a_{t,i}/\tau)}{\sum_{j=1}^{t} \exp(a_{t,j}/\tau)}, \qquad i = 1, \ldots, t.$$

Let

$$\bar{a}_t(\tau) := \sum_{j=1}^{t} \alpha_{t,j}(\tau) \, a_{t,j}$$

denote the average logit at position $t$ under the attention distribution $\alpha_t(\tau)$.

Then, for every $i \in \{1, \ldots, t\}$,

$$\frac{\partial}{\partial \tau} \alpha_{t,i}(\tau) = \frac{1}{\tau^2} \alpha_{t,i}(\tau) \left( \bar{a}_t(\tau) - a_{t,i} \right).$$

In particular:

- If $a_{t,i} > \bar{a}_t(\tau)$, then $\frac{\partial}{\partial \tau} \alpha_{t,i}(\tau) < 0$, so the attention on index $i$ *decreases* as $\tau$ increases.
- If $a_{t,i} < \bar{a}_t(\tau)$, then $\frac{\partial}{\partial \tau} \alpha_{t,i}(\tau) > 0$, so the attention on index $i$ *increases* as $\tau$ increases.
- If $a_{t,i} = \bar{a}_t(\tau)$, then $\frac{\partial}{\partial \tau} \alpha_{t,i}(\tau) = 0$.

**Proof.**

For a fixed $t$, write $a_i := a_{t,i}$ and $\alpha_i(\tau) := \alpha_{t,i}(\tau)$ for $i = 1, \ldots, t$ to lighten notation. By definition,

$$\alpha_i(\tau) = \frac{\exp(a_i/\tau)}{Z(\tau)}, \qquad Z(\tau) := \sum_{j=1}^{t} \exp(a_j/\tau).$$

We first differentiate the log-attention with respect to $\tau$:

$$\log \alpha_i(\tau) = \frac{a_i}{\tau} - \log Z(\tau).$$

Taking derivatives gives

$$\frac{\partial}{\partial \tau} \log \alpha_i(\tau) = -\frac{a_i}{\tau^2} - \frac{1}{Z(\tau)} \frac{\partial Z(\tau)}{\partial \tau}.$$

Next we compute $\frac{\partial Z(\tau)}{\partial \tau}$. By the chain rule,

$$\frac{\partial Z(\tau)}{\partial \tau} = \sum_{j=1}^{t} \frac{\partial}{\partial \tau} \exp(a_j/\tau) = \sum_{j=1}^{t} \exp(a_j/\tau) \left( -\frac{a_j}{\tau^2} \right) = -\frac{1}{\tau^2} \sum_{j=1}^{t} a_j \exp(a_j/\tau).$$

Thus

$$\frac{1}{Z(\tau)} \frac{\partial Z(\tau)}{\partial \tau} = -\frac{1}{\tau^2} \frac{\sum_{j=1}^{t} a_j \exp(a_j/\tau)}{\sum_{k=1}^{t} \exp(a_k/\tau)} = -\frac{1}{\tau^2} \sum_{j=1}^{t} \alpha_j(\tau) \, a_j = -\frac{1}{\tau^2} \bar{a}(\tau),$$

where

$$\bar{a}(\tau) := \sum_{j=1}^{t} \alpha_j(\tau) \, a_j$$

is the average logit under the current attention distribution.

Plugging this back into the derivative of $\log \alpha_i(\tau)$, we get

$$\frac{\partial}{\partial \tau} \log \alpha_i(\tau) = -\frac{a_i}{\tau^2} + \frac{1}{\tau^2} \bar{a}(\tau) = \frac{1}{\tau^2} \left( \bar{a}(\tau) - a_i \right).$$

Finally, we move from the derivative of the log-attention to the derivative of the attention itself. Since

$$\frac{\partial}{\partial \tau} \alpha_i(\tau) = \alpha_i(\tau) \frac{\partial}{\partial \tau} \log \alpha_i(\tau),$$

we obtain

$$\frac{\partial}{\partial \tau} \alpha_i(\tau) = \alpha_i(\tau) \frac{1}{\tau^2} \left( \bar{a}(\tau) - a_i \right) = \frac{1}{\tau^2} \alpha_i(\tau) \left( \bar{a}(\tau) - a_i \right),$$

which is the claimed formula.

The sign statements follow directly:

- If $a_i > \bar{a}(\tau)$, then $\bar{a}(\tau) - a_i < 0$, so $\frac{\partial}{\partial \tau} \alpha_i(\tau) < 0$ and the attention on $i$ decreases with $\tau$.
- If $a_i < \bar{a}(\tau)$, then $\bar{a}(\tau) - a_i > 0$, so $\frac{\partial}{\partial \tau} \alpha_i(\tau) > 0$ and the attention on $i$ increases with $\tau$.
- If $a_i = \bar{a}(\tau)$, then $\frac{\partial}{\partial \tau} \alpha_i(\tau) = 0$.

This shows that increasing the temperature shifts mass from positions with above-average logits to positions with below-average logits, which matches the intuitive picture of attention becoming flatter. $\qquad \square$

### A.6.2 LEMMA 2 (ATTENTION ENTROPY INCREASES BY INCREASING TEMPERATURE)

Fix a position $t$ and attention logits $a_{t,1}, \ldots, a_{t,t} \in \mathbb{R}$. For $\tau > 0$, define

$$\alpha_{t,i}(\tau) = \frac{\exp(a_{t,i}/\tau)}{\sum_{j=1}^{t} \exp(a_{t,j}/\tau)}, \qquad i = 1, \ldots, t,$$

and the entropy

$$H_t(\tau) := -\sum_{i=1}^{t} \alpha_{t,i}(\tau) \log \alpha_{t,i}(\tau).$$

Let

$$\bar{a}_t(\tau) := \sum_{j=1}^{t} \alpha_{t,j}(\tau) \, a_{t,j}$$

be the average logit at position $t$ under $\alpha_t(\tau)$. Then

$$\frac{\partial}{\partial \tau} H_t(\tau) = \frac{1}{\tau^3} \operatorname{Var}_{\alpha_t(\tau)} \big( a_{t,1}, \ldots, a_{t,t} \big) \ \geq \ 0,$$

with strict inequality whenever the logits $a_{t,1}, \ldots, a_{t,t}$ are not all equal.

**Proof.**

Fix $t$ and write $a_i := a_{t,i}$ and $\alpha_i(\tau) := \alpha_{t,i}(\tau)$ for $i = 1, \ldots, t$ to lighten notation. Let

$$Z(\tau) := \sum_{j=1}^{t} \exp(a_j/\tau), \qquad \bar{a}(\tau) := \sum_{j=1}^{t} \alpha_j(\tau) \, a_j.$$

By definition,

$$\alpha_i(\tau) = \frac{\exp(a_i/\tau)}{Z(\tau)}, \qquad \log \alpha_i(\tau) = \frac{a_i}{\tau} - \log Z(\tau).$$

Thus the entropy can be written as

$$H(\tau) := -\sum_{i=1}^{t} \alpha_i(\tau) \log \alpha_i(\tau) = -\sum_{i=1}^{t} \alpha_i(\tau) \Big( \frac{a_i}{\tau} - \log Z(\tau) \Big).$$

Using $\sum_i \alpha_i(\tau) = 1$, this simplifies to

$$H(\tau) = -\frac{1}{\tau} \sum_{i=1}^{t} \alpha_i(\tau) a_i + \log Z(\tau) = -\frac{1}{\tau} \bar{a}(\tau) + \log Z(\tau).$$

Differentiate $H(\tau)$ with respect to $\tau$:

$$\frac{\partial H}{\partial \tau} = \frac{1}{\tau^2}\,\bar{a}(\tau) - \frac{1}{\tau}\,\frac{\partial \bar{a}(\tau)}{\partial \tau} + \frac{1}{Z(\tau)}\,\frac{\partial Z(\tau)}{\partial \tau}.$$

We compute $\frac{\partial Z(\tau)}{\partial \tau}$:

$$\frac{\partial Z(\tau)}{\partial \tau} = \sum_{j=1}^{t} \frac{\partial}{\partial \tau}\exp(a_j/\tau) = \sum_{j=1}^{t}\exp(a_j/\tau)\Big(-\frac{a_j}{\tau^2}\Big) = -\frac{1}{\tau^2}\sum_{j=1}^{t} a_j \exp(a_j/\tau).$$

Hence

$$\frac{1}{Z(\tau)}\frac{\partial Z(\tau)}{\partial \tau} = -\frac{1}{\tau^2}\frac{\sum_{j=1}^{t} a_j \exp(a_j/\tau)}{\sum_{k=1}^{t}\exp(a_k/\tau)} = -\frac{1}{\tau^2}\sum_{j=1}^{t}\alpha_j(\tau)\,a_j = -\frac{1}{\tau^2}\,\bar{a}(\tau).$$

Plugging this into the expression for $\partial H/\partial \tau$ gives

$$\frac{\partial H}{\partial \tau} = \frac{1}{\tau^2}\,\bar{a}(\tau) - \frac{1}{\tau}\,\frac{\partial \bar{a}(\tau)}{\partial \tau} - \frac{1}{\tau^2}\,\bar{a}(\tau) = -\frac{1}{\tau}\,\frac{\partial \bar{a}(\tau)}{\partial \tau}.$$

We now compute $\frac{\partial \bar{a}(\tau)}{\partial \tau}$. By definition,

$$\bar{a}(\tau) = \sum_{i=1}^{t}\alpha_i(\tau)\,a_i, \quad \text{so} \quad \frac{\partial \bar{a}(\tau)}{\partial \tau} = \sum_{i=1}^{t} a_i\,\frac{\partial \alpha_i(\tau)}{\partial \tau}.$$

From Lemma 1 we have, for each $i$,

$$\frac{\partial \alpha_i(\tau)}{\partial \tau} = \frac{1}{\tau^2}\,\alpha_i(\tau)\,\big(\bar{a}(\tau) - a_i\big).$$

Therefore,

$$\frac{\partial \bar{a}(\tau)}{\partial \tau} = \frac{1}{\tau^2}\sum_{i=1}^{t} a_i\,\alpha_i(\tau)\,\big(\bar{a}(\tau) - a_i\big) = \frac{1}{\tau^2}\Big(\bar{a}(\tau)\sum_{i=1}^{t}\alpha_i(\tau)a_i - \sum_{i=1}^{t}\alpha_i(\tau)a_i^2\Big).$$

Since $\sum_i \alpha_i(\tau)a_i = \bar{a}(\tau)$, this becomes

$$\frac{\partial \bar{a}(\tau)}{\partial \tau} = \frac{1}{\tau^2}\Big(\bar{a}(\tau)^2 - \sum_{i=1}^{t}\alpha_i(\tau)a_i^2\Big) = -\frac{1}{\tau^2}\Big(\sum_{i=1}^{t}\alpha_i(\tau)a_i^2 - \bar{a}(\tau)^2\Big).$$

The term in parentheses is the variance of the logits under $\alpha(\tau)$:

$$\mathrm{Var}_{\alpha(\tau)}(a) := \sum_{i=1}^{t}\alpha_i(\tau)a_i^2 - \bar{a}(\tau)^2.$$

Hence

$$\frac{\partial \bar{a}(\tau)}{\partial \tau} = -\frac{1}{\tau^2}\,\mathrm{Var}_{\alpha(\tau)}(a).$$

Substituting into $\partial H/\partial \tau$ yields

$$\frac{\partial H}{\partial \tau} = -\frac{1}{\tau}\Big(-\frac{1}{\tau^2}\,\mathrm{Var}_{\alpha(\tau)}(a)\Big) = \frac{1}{\tau^3}\,\mathrm{Var}_{\alpha(\tau)}(a).$$

Since variance is always non-negative and equals zero only when all logits $a_1, \ldots, a_t$ are equal, we obtain

$$\frac{\partial H}{\partial \tau} \geq 0,$$

with strict inequality whenever $a_1, \ldots, a_t$ are not all equal. This proves that the attention entropy increases with temperature unless the attention is already uniform. $\qquad\square$

### A.6.3 LEMMA 3 (EFFECT ON FACTUAL-TOKEN LOGITS)

Fix a factual position $t \in F$. Under Assumptions A1 and A3, the logit margin of the factual token decreases as $\tau$ increases above 1, and becomes negative for sufficiently large $\tau$. More precisely, there exists $\tau_F \geq 1$ such that for all $\tau \geq \tau_F$,

$$\ell_t(y_t^\star; \theta_\tau) - \max_{u \neq y_t^\star} \ell_t(u; \theta_\tau) \leq -\frac{m_F}{2} < 0.$$

**Proof.**

By A1, most of the base attention mass at position $t$ lies on the set $S_t$, whose value vectors strengthen the logit of $y_t^\star$. The base attention output can be written as

$$z_t = \sum_{i=1}^{t} \alpha_{t,i} v_i = \sum_{i \in S_t} \alpha_{t,i} v_i + \sum_{i \notin S_t} \alpha_{t,i} v_i.$$

For large $\tau$, Lemma 2 implies that $\alpha_{t,i}(\tau)$ approaches the uniform distribution as $\tau \to \infty$. Hence

$$z_t(\tau) = \sum_{i=1}^{t} \alpha_{t,i}(\tau) v_i \xrightarrow[\tau \to \infty]{} \frac{1}{t} \sum_{i=1}^{t} v_i = \lambda_S z_t(S_t) + \lambda_{\bar{S}_t} z_t(\bar{S}_t),$$

for weights $\lambda_S, \lambda_{\bar{S}}$ determined by $|S_t|$ and $t$. Define the change

$$\Delta z_t(\tau) := z_t(\tau) - z_t.$$

By Lemma 1, as $\tau$ increases, $\Delta z_t(\tau)$ moves the attention output away from the sharp pattern that favors $S_t$, and toward the $\bar{S}_t$ with attention weights lower than entropy.

Let $u \neq y_t^\star$ be any competing token. The logit difference at temperature $\tau$ is

$$\ell_t(y_t^\star; \theta_\tau) - \ell_t(u; \theta_\tau) = \langle W_{y_t^\star} - W_u, \, z_t(\tau) \rangle + (b_{y_t^\star} - b_u).$$

Subtracting the difference at $\tau = 1$ gives

$$\Delta_\tau = \langle W_{y_t^\star} - W_u, \, z_t(\tau) - z_t \rangle.$$

By A1, putting more weight on $S_t$ increases the factual margin, so moving away from $S_t$ (as smoothing does) decreases it. Thus, for large enough $\tau$, the inner product above is negative and can be bounded above by a negative constant once $\alpha_t(\tau)$ is close to uniform.

By A3, at $\tau = 1$ the factual margin is positive:

$$\ell_t(y_t^\star; \theta) - \ell_t(u; \theta) \geq \Delta_F > 0.$$

By continuity in $\tau$ (A4), the margin decreases continuously as the attention pattern is smoothed. Since the margin becomes negative for large $\tau$, the intermediate value theorem guarantees a point $\tau_F$ where it crosses zero. For any $\tau \geq \tau_F$, the margin is strictly negative, and by adjusting the threshold we may ensure the bound $-m_F/2$.

This implies that for $\tau \geq \tau_F$,

$$p_{\theta_\tau}(y_t^\star \mid x \circ y_{<t}) \leq \frac{1}{1 + \exp(m_F/2)} =: \epsilon_F < \frac{1}{2},$$

so the factual token is no longer the most likely output. $\square$

### A.6.4 LEMMA 4 (EFFECT ON FUNCTION-TOKEN LOGITS IS SMALL)

Fix a function position $t \in G$. Under Assumptions A2 and A4, for any $\eta > 0$ there exists $\bar{\tau}_G \geq 1$ such that for all $\tau \in [1, \bar{\tau}_G]$,

$$\left| \ell_t(y_t; \theta_\tau) - \ell_t(y_t; \theta) \right| \leq \eta.$$

**Proof.**

For any compact interval $[1, \bar{\tau}_G]$, the attention weights $\alpha_t(\tau)$ vary continuously in $\tau$ and stay inside the simplex. Therefore $z_t(\tau)$ is a continuous function of $\tau$.

By A2, the logit of the correct function token is $L$-Lipschitz in the attention output:

$$\left| \ell_t(y_t; \theta_\tau) - \ell_t(y_t; \theta) \right| \leq L \left\| z_t(\tau) - z_t \right\|_2.$$

Since $z_t(\tau) \to z_t$ as $\tau \to 1$ (by A4 and continuity of the attention map), for any $\eta > 0$ we can choose $\bar{\tau}_G > 1$ so that

$$\| z_t(\tau) - z_t \|_2 \leq \eta/L \qquad \text{for all } \tau \in [1, \bar{\tau}_G].$$

Substituting into the Lipschitz bound yields

$$\left| \ell_t(y_t; \theta_\tau) - \ell_t(y_t; \theta) \right| \leq \eta,$$

which proves the claim. $\qquad\square$

### A.6.5 LEMMA 5 (SMALL LOGIT CHANGE IMPLIES SMALL CROSS-ENTROPY CHANGE)

Let $p$ and $q$ be two distributions over $V$ whose logits differ at the true token by at most $\eta$. Then the increase in negative log-likelihood at that token is at most a function $c(\eta)$ with $c(\eta) \to 0$ as $\eta \to 0$.

Formally, let $\ell_\theta$ and $\ell_{\theta_\tau}$ be two logit vectors. If for some token $w$,

$$\left| \ell_{\theta_\tau}(w) - \ell_\theta(w) \right| \leq \eta,$$

and the remaining logit differences are uniformly bounded, then

$$- \log p_{\theta_\tau}(w) \ \leq \ - \log p_\theta(w) + c(\eta).$$

**Proof sketch.** The softmax map from logits to probabilities is smooth and Lipschitz on any compact region of logit space, and the negative log-probability of a fixed token is smooth as well. Thus a small change in the logits produces a small change in the negative log-likelihood. The function $c(\eta)$ follows from the Lipschitz constants of the softmax and the log operation. $\qquad\square$

Combining Lemma 4 and Lemma 5, for any tolerance $\delta_G > 0$ we may choose $\bar{\tau}_G > 1$ so that for all $\tau \in [1, \bar{\tau}_G]$ and all function tokens $t \in G$,

$$- \log p_{\theta_\tau}(y_t \mid x \circ y_{<t}) \leq - \log p_\theta(y_t \mid x \circ y_{<t}) + \delta_G.$$

### A.7 MAIN THEOREM

**Theorem: Attention smoothing yields forgetting with fluency.**

Assume A1–A4 hold for all forget examples $(x, y) \in D_F$ and for their factual and function positions. Then there exists a temperature interval $[\tau_0, \tau_1]$ with

$$1 < \tau_0 \leq \tau_1 < \infty$$

such that:

- **Forgetting:** For all $\tau \in [\tau_0, \tau_1]$, the smoothed model $\theta_\tau$ forgets the factual tokens in $D_F$. In particular, for every factual position $t \in F$,

$$p_{\theta_\tau}(y_t^\star \mid x \circ y_{<t}) \leq \epsilon_F,$$

  and the average factual negative log-likelihood is at least $L_F > 0$.

- **Fluency:** For all $\tau \in [\tau_0, \tau_1]$, the increase in average loss on function tokens (over both $D_F$ and $D_R$) is at most $\delta_G$:

$$\frac{1}{|G|} \sum_{t \in G} - \log p_{\theta_\tau}(y_t \mid x \circ y_{<t}) \leq \frac{1}{|G|} \sum_{t \in G} - \log p_\theta(y_t \mid x \circ y_{<t}) + \delta_G.$$

Thus there is a non-trivial range of temperatures where factual knowledge is forgotten while fluent language behavior is preserved.

**Proof.**

*Step 1: Forgetting at sufficiently large $\tau$.* For each factual position $t \in F$, Lemma 3 provides a temperature $\tau_F(t)$ such that for all $\tau \geq \tau_F(t)$,

$$\ell_t(y_t^\star; \theta_\tau) - \max_{u \neq y_t^\star} \ell_t(u; \theta_\tau) \leq -\frac{m_F}{2}.$$

Hence

$$p_{\theta_\tau}(y_t^\star \mid x \circ y_{<t}) \leq \epsilon_F,$$

for $\epsilon_F < 1/2$ depending only on $m_F$.

Define

$$\tau_F := \max_{(x,y) \in D_F} \max_{t \in F} \tau_F(t).$$

For all $\tau \geq \tau_F$, the forgetting inequality holds for every factual position in every forget example, and the average factual loss is at least $L_F = -\log \epsilon_F$.

*Step 2: Fluency at sufficiently small $\tau$.* For each function position $t \in G$, Lemma 4 states that for any $\eta > 0$ there exists $\bar{\tau}_G(t) > 1$ such that for all $\tau \in [1, \bar{\tau}_G(t)]$,

$$\left| \ell_t(y_t; \theta_\tau) - \ell_t(y_t; \theta) \right| \leq \eta.$$

Lemma 5 then ensures that the extra loss on each function token is at most $c(\eta)$, where $c(\eta) \to 0$ as $\eta \to 0$.

Choose $\eta$ so that $c(\eta) \leq \delta_G$, and define

$$\bar{\tau}_G := \min_{(x,y)} \min_{t \in G} \bar{\tau}_G(t).$$

For all $\tau \in [1, \bar{\tau}_G]$,

$$-\log p_{\theta_\tau}(y_t \mid x \circ y_{<t}) \leq -\log p_\theta(y_t \mid x \circ y_{<t}) + \delta_G,$$

and averaging this over all function tokens gives the bound in the theorem.

*Step 3: Establishing a common temperature range.* We have:

- Forgetting holds for all $\tau \geq \tau_F$.
- Fluency holds for all $\tau \in [1, \bar{\tau}_G]$.

Both hold simultaneously for all

$$\tau \in [\tau_F, \bar{\tau}_G],$$

which is non-empty whenever $\tau_F \leq \bar{\tau}_G$.

This condition reflects the structure in A1–A3: factual tokens depend on precise attention patterns that collapse quickly when smoothed, while function tokens depend on broader patterns that remain stable under mild smoothing.

Choose any $\tau_0, \tau_1$ satisfying

$$1 < \tau_F \leq \tau_0 \leq \tau_1 \leq \bar{\tau}_G < \infty.$$

Then for all $\tau \in [\tau_0, \tau_1]$, both forgetting and fluency hold. $\qquad\square$

# B BASELINES

**Notation.** Let $P(y \mid x; \theta)$ denote the probability of an output sequence $y = (y_1, \ldots, y_T)$ given input $x$ under a model parameterized by $\theta$. This probability is defined as:

$$P(y \mid x; \theta) = \prod_{t=1}^{T} p(y_t \mid x \circ y_{<t}; \theta)^{\frac{1}{T}} .$$

**Forget Loss.** Existing methods can be broadly categorized into *Convergence-based Unlearning* and *Divergence-based Unlearning*. The baselines we use are:

- **Gradient Ascent (GA)** (Yao et al., 2023) maximizes the prediction loss on the forget set, effectively reversing the training objective:

$$\mathcal{L}_{\mathrm{GA}}(\mathcal{D}_{\mathrm{F}}; \theta) = -\mathbb{E}_{(x,y) \sim \mathcal{D}_{\mathrm{F}}} \left[ \frac{1}{T} \sum_{t=1}^{T} -\log p(y_t \mid x \circ y_{<t}; \theta) \right] . \tag{6}$$

- **Negative Preference Optimization (NPO)** (Zhang et al., 2024b) is derived from Direct Preference Optimization (DPO) (Rafailov et al., 2023). It treats forget-set answers as negative samples while omitting positive terms:

$$\mathcal{L}_{\mathrm{NPO}}(\mathcal{D}_{\mathrm{F}}; \theta) = -\frac{2}{\beta} \mathbb{E}_{(x,y) \sim \mathcal{D}_{\mathrm{F}}} \left[ \log \sigma \left( -\beta \log \frac{P(y \mid x; \theta)}{P(y \mid x; \theta_{\mathrm{Base}})} \right) \right] , \tag{7}$$

where $\sigma(t) = 1/(1 + e^{-t})$, $\beta$ is a hyperparameter, and $\theta_{\mathrm{ref}}$ is the fixed reference model. NPO can be viewed as GA with adaptive gradient scaling (Zhang et al., 2024b).

- **Maximizing Entropy (ME)** (Yuan et al., 2024) minimize the KL divergence between the predicted distribution for each token and a uniform distribution with vocabulary size.

$$\mathcal{L}_{\mathrm{ME}}(\mathcal{D}_{\mathrm{F}}; \theta) = \mathbb{E}_{(x,y) \sim \mathcal{D}_{\mathrm{F}}} \left[ \frac{1}{T} \sum_{t=1}^{T} \mathrm{KL}\big(\mathcal{U}_{[K]} || p(\cdot \mid x \circ y_{<t}; \theta)\big) \right] , \tag{8}$$

where $\mathcal{U}_{[K]}$ is a uniform distribution over the vocabulary of size $K$, where each value is $1/K$.

- **IDK Fine-tune (IDK)** (Maini et al., 2024) reframes unlearning as instruction tuning by relabeling forget-set questions with random responses from $\mathcal{D}_{\mathrm{IDK}}$, a pool of rejection templates (e.g., "Sorry, I don't know."). Its loss is

$$\mathcal{L}_{\mathrm{IDK}}(\mathcal{D}_{\mathrm{F}}, \mathcal{D}_{\mathrm{IDK}}; \theta) = \mathbb{E}_{x \sim \mathcal{D}_{\mathrm{F}}, y \sim \mathcal{D}_{\mathrm{IDK}}} \left[ -\log P(y \mid x; \theta) \right] . \tag{9}$$

- **Direct Preference Optimization (DPO)** (Zhang et al., 2024b) applies the standard DPO loss (Rafailov et al., 2023), using forget-set answers as negatives and rejection templates from $\mathcal{D}_{\mathrm{IDK}}$ as positives.

$$\mathcal{L}_{\mathrm{DPO}}(\mathcal{D}_{\mathrm{F}}, \mathcal{D}_{\mathrm{IDK}}; \theta; \theta_{\mathrm{ref}}) = -\frac{1}{\beta} \mathbb{E}_{(x,y) \sim \mathcal{D}_{\mathrm{F}}, y' \sim \mathcal{D}_{\mathrm{IDK}}}$$
$$\left[ \log \sigma \left( \beta \log \frac{P(y' \mid x; \theta)}{P(y' \mid x; \theta_{\mathrm{base}})} - \beta \log \frac{P(y \mid x; \theta)}{P(y \mid x; \theta_{\mathrm{base}})} \right) \right] , \tag{10}$$

where $\theta_{\mathrm{base}}$ denotes the parameter of the reference model, which is the initial base model for unlearning.

- **SimNPO** (Fan et al., 2024). It derives from NPO, whose reward function is given by the comparison with the reference model. In contrast, SimNPO takes a reference-free but length-normalized reward formulation, so they can mitigate the reference model bias in NPO by replacing its reward formulation, as follows:

$$\mathcal{L}_{\mathrm{SimNPO}}(\mathcal{D}_{\mathrm{F}}; \theta) = -\frac{2}{\beta} \mathbb{E}_{(x,y) \sim \mathcal{D}_{\mathrm{F}}} \left[ \log \sigma \left( -\frac{\beta}{|y|} \log P(y \mid x; \theta) - \gamma \right) \right] , \tag{11}$$

where $\gamma \geq 0$ is the reward margin parameter, inherited from SimPO, which defines the margin of preference for a desired response over a dispreferred one.

- **Representation Misdirection (RMU)** (Li et al., 2024) misdirects internal representations on the forget set by pushing layer-$\ell$ activations toward a fixed random direction with amplified norm, corrupting downstream processing. It's forget loss is

$$\mathcal{L}_{\text{RMU}} = \mathbb{E}_{x \sim \mathcal{D}_{\text{F}}} \left[ \frac{1}{T} \sum_{t=1}^{T} \left\| H^\ell(x_{<t}; \theta) - c \cdot u \right\|_2^2 \right], \tag{12}$$

where $H^\ell(x_{<t}; \theta)$ denotes the hidden state at layer $\ell$ of the model parameterized by $\theta$, given the prefix $x_{<t}$, $u$ is a random unit vector, $c > 0$ is a scaling constant, and $T$ is the sequence length of $x$.

IDK and DPO are only applicable in QA-style datasets, since they require rejection templates as positive samples.

**Retain Loss.** In addition to the GD and KL regularization losses introduced in Section 2.3, we further include the Answer Preservation (AP) and Mean Squared Error (MSE) loss as an additional baseline component.

- **Answer Preservation (AP)**. To prevent unlearned models from becoming overly ignorant during targeted unlearning, (Yuan et al., 2024) proposed the Answer Preservation (AP) loss as a regularization term. Unlike standard GD or KL regularization, AP explicitly balances two objectives on the retain set: (1) reducing the probability of rejection templates, and (2) maintaining the probability of the original answers. Formally, the AP loss is defined as:

$$\mathcal{L}_{\text{AP}}(\mathcal{D}_{\text{R}}, \mathcal{D}_{\text{IDK}}; \theta) = -\frac{1}{\beta} \mathbb{E}_{(x,y) \sim \mathcal{D}_{\text{R}}, y' \sim \mathcal{D}_{\text{IDK}}} \left[ \log \sigma(-\beta \log \frac{P(y' \mid x; \theta)}{P(y \mid x; \theta)}) \right], \tag{13}$$

where $\sigma(\cdot)$ is the sigmoid function and $\beta$ is a temperature parameter.

- **Mean Squared Error (MSE)** (Li et al., 2024). The motivation of this loss is to limit the degradation of general capabilities by explicitly constraining the updated model's internal representations to remain close to those of the base model. Concretely, given the retain dataset $\mathcal{D}_{\text{R}}$, we impose an $\ell^2$ penalty between the hidden activations of the updated model and the base model:

$$\mathcal{L}_{\text{MSE}}(\mathcal{D}_{\text{R}}; \theta) = \mathbb{E}_{x \sim \mathcal{D}_{\text{R}}} \left[ \frac{1}{T} \sum_{t=1}^{T} \left\| H^\ell(x_{<t}; \theta) - H^\ell(x_{<t}; \theta_{\text{base}}) \right\|_2^2 \right], \tag{14}$$

where $H^\ell(x_{<t}; \theta)$ denotes the hidden state at layer $\ell$ of the model parameterized by $\theta$, given the prefix $x_{<t}$, and $T$ is the number of tokens in $x$. This loss explicitly encourages the updated model to preserve activation-level similarity with the reference model on the retain set, thereby mitigating the risk of excessive utility loss during unlearning.

# C  EVALUATION METRICS

## C.1  RIGHT TO BE FORGOTTEN

**Notation.** Let $g(x; \theta)$ denote the decoded output produced by a model parameterized by $\theta$ for input $x$.

**Metrics.** We evaluate the Right-to-be-Forgotten scenario using the following metrics:

- **ROUGE (R)** We use ROUGE-L recall (Maini et al., 2024) to compare the model's decoded output $g(x; \theta)$ with the ground truth answer $y$. The score, denoted as $\text{ROUGE}(g(x; \theta), y)$, captures the longest common subsequence overlap at the word level.
- **Probability (P)** We measure the model's likelihood of producing the ground-truth answer $y$ (Maini et al., 2024). For a question–answer pair $(x, y)$, we compute the normalized conditional probability:

$$P(y \mid x; \theta) = \prod_{t=1}^{T} p(y_t \mid x \circ y_{<t}; \theta)^{\frac{1}{T}},$$

where $T$ is the answer length, $y_t$ is the $t$-th token, and $y_{<t}$ denotes the prefix up to position $t$.

- **Truth Ratio (TR)** We assess whether the model assigns higher likelihood to correct answers than to incorrect ones (Maini et al., 2024; Yuan et al., 2024). The metric TR compares the average normalized conditional probability of perturbed answers $\hat{y}$, which are plausible but incorrect variants of $y$, against that of a paraphrased answer $\tilde{y}$, which is a valid rephrasing of $y$. Formally,

$$\mathrm{TR}(y \mid x; \theta) \;=\; \frac{\frac{1}{|\hat{y}|} \sum_{i=1}^{|\hat{y}|} P(\hat{y}_i \mid x; \theta)}{P(\tilde{y} \mid x; \theta)}.$$

A model lacking relevant knowledge should assign similar probabilities to correct and incorrect answers. For evaluation, we report $\max(0, 1 - \mathrm{TR})$ on the retain set and $1 - \min(\mathrm{TR}, 1/\mathrm{TR})$ on the forget set.

- **Token Entropy (TE)** We evaluate the lexical diversity of the model's output (Yuan et al., 2024). Some unlearned models often generate long, repetitive continuations (e.g., gibberish output) that reduce readability. To quantify this effect, we compute a normalized token entropy:

$$\mathrm{TE}(g(x; \theta_u)) \;=\; \frac{-\sum_{i=1}^{m} f(w_i) \log_2 f(w_i)}{\log_2 |g(x; \theta)|},$$

where $|g(x; \theta)|$ is the output length, $m$ is the number of unique tokens, and $f(w_i)$ denotes the frequency of token $w_i$. Lower TE indicates excessive repetition and incoherent outputs, while higher TE reflects more diverse and readable generations.

- **Cosine Similarity (CS)** We measure the semantic similarity between the model's output before and after unlearning on the retain set (Yuan et al., 2024). In line with the semantic textual similarity task (Cer et al., 2017), we use Sentence-BERT (Reimers & Gurevych, 2019) to embed the output produced by the base model and the output produced by the unlearned model, and then compute their cosine similarity, truncated at zero:

$$\max\big(\mathrm{Cos}\big(g(x; \theta_{base}), \, g(x; \theta)\big), \, 0\big).$$

This metric captures semantic drift: even if surface overlap (e.g., ROUGE) remains high, cosine similarity decreases when the unlearned model appends irrelevant or fabricated content.

- **Entailment Score (ES)** We assess the factual consistency of model outputs with respect to ground-truth answers using textual entailment (Natural Language Inference, NLI) (Yuan et al., 2024). NLI evaluates whether a premise entails, contradicts, or is neutral with respect to a hypothesis, and has been widely applied in NLP evaluation (Poliak, 2020). Formally, a text $t$ entails a hypothesis $h$ ($t \Rightarrow h$) if a human reading $t$ would reasonably infer $h$ to be true.

  We use a pre-trained NLI model (Sileo, 2023) to predict the relationship between each model output and its ground-truth answer (Liu et al., 2024b). The entailment score is defined as the proportion of predictions labeled as "entailment", which should be higher on the retain set and lower on the forget set.

## C.2 COPYRIGHT SCENARIO

We evaluate the copyright scenario (MUSE tasks) using the following metrics:

- **Verbatim Memorization (VerbMem)** We assess whether the model reproduces training data verbatim (Shi et al., 2024). Given a forget-set sequence $x \in \mathcal{D}_{\mathrm{F}}$, we provide the model $g$ with the first $l$ tokens $x_{[:l]}$ and compare its continuation with the ground truth suffix $x_{[l+1:]}$ using the ROUGE-L F1 score. The metric is averaged over all examples in $\mathcal{D}_{\mathrm{F}}$:

$$\mathrm{VerbMem}(\theta, \mathcal{D}_{\mathrm{F}}) = \frac{1}{|\mathcal{D}_{\mathrm{F}}|} \sum_{x \in \mathcal{D}_{\mathrm{F}}} \mathrm{ROUGE}(g(x_{\leq l}; \theta), x_{>l}).$$

A lower VerbMem indicates stronger protection against verbatim leakage.

- **Knowledge Memorization (KnowMem)** We measure whether the model retains factual knowledge of the forget set (Shi et al., 2024). For each sample $(x, y) \in \mathcal{D}_F$, we query the model with $x$ and compare its answer $g(x; \theta)$ with the ground truth $y$ using ROUGE. The metric is averaged over all pairs:

$$\text{KnowMem}(\theta, \mathcal{D}_F) = \frac{1}{|\mathcal{D}_F|} \sum_{(x,y) \in \mathcal{D}_F} \text{ROUGE}(g(x; \theta), y).$$

A lower KnowMem reflects more effective removal of copyrighted or sensitive knowledge.

- **Privacy Leakage (PrivLeak)** To evaluate privacy preservation, we follow (Shi et al., 2024), and adopt the state-of-the-art Min-$K\%$ Prob method (Shi et al., 2023) and compute the AUC-ROC score (Murakonda et al., 2021; Shokri et al., 2017) for discriminating $\mathcal{D}_F$ from a holdout set $\mathcal{D}_{\text{holdout}}$. The privacy leakage is then defined relative to a retrained model:

$$\text{PrivLeak} = \frac{\text{AUC}(\theta; \mathcal{D}_F, \mathcal{D}_{\text{holdout}}) - \text{AUC}(\theta_{\text{retrain}}; \mathcal{D}_F, \mathcal{D}_{\text{holdout}})}{\text{AUC}(\theta_{\text{retrain}}; \mathcal{D}_F, \mathcal{D}_{\text{holdout}})}.$$

A good unlearning algorithm yields PrivLeak close to zero, while large positive or negative values indicate over- or under-unlearning.

## D    CONTINUAL UNLEARNING SCENARIO

Figures 5 and 6 report FE and MU for continual unlearning on TOFU. DPO attains higher FE than ASU but drives MU to 0.0, indicating extreme ignorance. ME achieves MU comparable to ASU, but ASU delivers higher FE, yielding a better average performance overall (as shown in Figure 3).

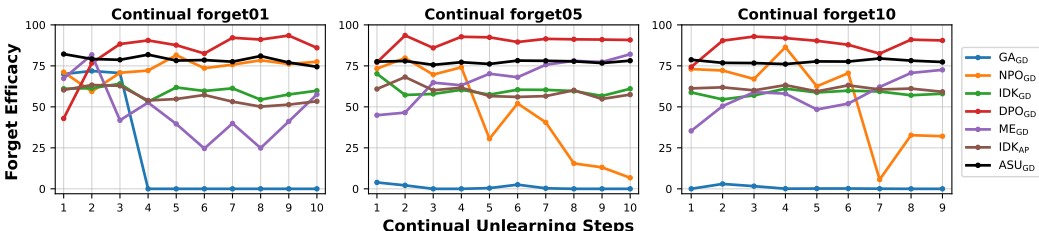

Figure 5: Forget Efficacy in continual forget01, forget05 and forget10 unlearning tasks.

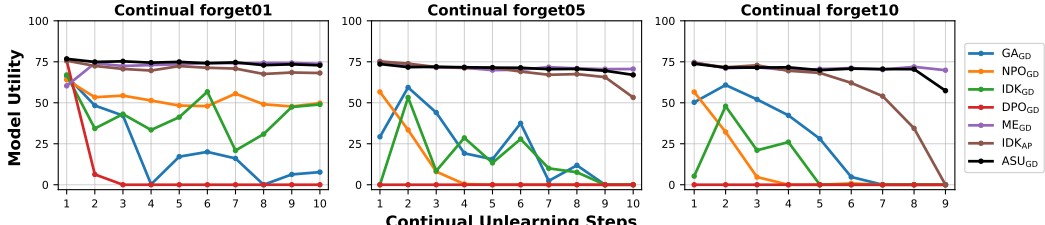

Figure 6: Model Utility in continual forget01, forget05 and forget10 unlearning tasks.

## E    REAL-WORLD UNLEARNING SCENARIO

Table 6 presents the detailed results for each metric in the real-world unlearning scenario, corresponding to the summary provided in Table 2.

## F    HAZARDOUS-KNOWLEDGE UNLEARNING SCENARIO

In addition to output-level alignment, we also match internal representations. We minimize the mean squared error (MSE) between hidden states of the model parameterized by $\theta$ and those of the

Table 6: **Detailed results of each metric in real-world unlearning scenario.**

| Method | Neighbor Set | | | | | | | Forget Set | | | | | |
|---|---|---|---|---|---|---|---|---|---|---|---|---|---|
| | R ↑ | P ↑ | TR ↑ | TE ↑ | CS ↑ | ES ↑ | MU ↑ | R ↓ | P ↓ | TR ↓ | TE ↑ | ES ↓ | FE ↑ |
| Base | 78.21 | 33.75 | 56.17 | 88.50 | 98.32 | 62.25 | 61.38 | 80.67 | 38.97 | 60.70 | 89.58 | 67.75 | 36.83 |
| **Divergence-based Unlearning** | | | | | | | | | | | | | |
| GA$_{GD}$ | 63.53 | 5.01 | 78.18 | 83.08 | 70.38 | 46.75 | 21.76 | 0.00 | 0.00 | 48.81 | 37.68 | 0.00 | 65.73 |
| GA$_{KL}$ | 51.77 | 26.69 | 62.03 | 72.80 | 64.50 | 28.50 | 43.72 | 0.00 | 0.00 | 69.94 | 0.00 | 0.00 | 0.00 |
| NPO$_{GD}$ | 50.41 | 8.71 | 42.84 | 69.39 | 57.80 | 11.00 | 21.38 | 42.28 | 5.93 | 39.31 | 66.41 | 4.75 | 71.44 |
| NPO$_{KL}$ | 50.55 | 17.51 | 43.05 | 68.79 | 55.38 | 11.50 | 27.32 | 41.27 | 9.22 | 38.20 | 67.53 | 3.00 | 72.11 |
| **Convergence-based Unlearning** | | | | | | | | | | | | | |
| DPO$_{GD}$ | 0.45 | 25.22 | 35.88 | 71.09 | 5.15 | 0.00 | 0.00 | 0.30 | 21.41 | 34.82 | 79.70 | 0.00 | 82.45 |
| DPO$_{KL}$ | 3.05 | 35.60 | 40.45 | 99.69 | 9.72 | 0.75 | 3.28 | 0.82 | 28.14 | 37.07 | 99.97 | 0.00 | 83.48 |
| IDK$_{GD}$ | 2.61 | 32.12 | 46.88 | 100.00 | 8.77 | 0.00 | 0.00 | 2.63 | 31.57 | 47.07 | 100.00 | 0.00 | 78.40 |
| IDK$_{AP}$ | 70.81 | 29.93 | 53.43 | 86.66 | 80.58 | 42.50 | 52.76 | 3.45 | 22.58 | 51.39 | 99.27 | 1.50 | 78.04 |
| ME$_{GD}$ | 70.25 | 21.21 | 58.12 | 90.66 | 82.57 | 42.75 | 47.96 | 2.43 | 0.19 | 22.65 | 16.46 | 0.25 | 48.10 |
| ASU$_{GD}$ | 69.10 | 37.30 | 46.55 | 85.08 | 80.36 | 41.75 | 54.10 | 33.30 | 13.37 | 31.25 | 73.84 | 3.25 | 76.97 |
| ASU$_{KL}$ | 69.96 | 42.97 | 44.29 | 88.91 | 82.56 | 41.50 | 55.76 | 30.32 | 19.74 | 31.05 | 91.38 | 5.25 | 79.60 |

attention-smoothed model $\theta_\tau$ at a chosen layer. Concretely, we align $\theta$ with $\theta_{\text{base}}$ on the retain set 14 and with $\theta_\tau$ on the forget set 15, as follows:

$$\mathcal{L}_{\text{ASU}(\ell)}(D_F; \theta; \theta_\tau) = \mathbb{E}_{x \sim \mathcal{D}_F} \left[ \frac{1}{|x|} \sum_{t=1}^{|x|} \left\| H^\ell(x_{<t}; \theta) - H^\ell(x_{<t}, \theta_\tau) \right\|_2^2 \right], \quad (15)$$

where $H^\ell(x_{<t}; \theta)$ denotes the hidden state at layer $\ell$ of the model parameterized by $\theta$, given the prefix $x_{<t}$.

**Setup.** We assess hazardous-knowledge removal using WMDP (Li et al., 2024). The forget set $D_f$ comprises WMDP-Biology and WMDP-Cyber corpora, and the retain set $D_r$ is Wikitext (Merity et al., 2017). Unlearned models are evaluated on the WMDP multiple-choice QA benchmark (zero-shot; select the option with highest conditional probability) to measure residual hazardous knowledge, and on MMLU (Hendrycks et al.) to measure general utility. We choose layer $\ell(7)$ as the unlearning layer, and we only update the MLP layers of three layers $\ell, \ell-1, \ell-2$ (7,6,5), which can be leveraged to save memory and efficiently unlearn on larger LMs (Li et al., 2024).

**Models.** We evaluate hazardous-knowledge removal on the following LLMs: Zephyr-7B-$\beta$ (Tunstall et al., 2023), Mistral-7B-Instruct-v0.2 (Jiang et al., 2023).

Table 7: Comparing base models and unlearning methods on question-answer evaluation (WMDP, MMLU). All WMDP and MMLU scores are percentage points.

**Baselines.** We compare against RMU (Li et al., 2024), SCRUB (Kurmanji et al., 2023), SSD (Foster et al., 2024), and LLMU (Yao et al., 2024b). Baseline runs are conducted on Zephyr-7B; in preliminary screening on this backbone, all baselines except RMU significantly affect Model Utility while not achieving good forget efficacy, so we do not extend them to the other models.

| Model | Method | WMDP (↓) | | MMLU (↑) |
|---|---|---|---|---|
| | | Bio | Cyber | |
| | Base | 64.3 | 44.8 | 58.5 |
| Zephyr-7B-$\beta$ | LLMU | 59.5 | 39.5 | 44.7 |
| | SCRUB | 43.8 | 39.3 | 51.2 |
| | SSD | 50.2 | 35.0 | 40.7 |
| | RMU | **31.2** | **28.2** | 57.0 |
| | ASU | 32.1 | 31.7 | **57.5** |
| | Base | 65.1 | 41.5 | 59.0 |
| Mistral-7B | RMU | **30.7** | 32.3 | **57.7** |
| | ASU | 31.5 | **29.5** | 57.2 |

**Performance on WMDP.** Table 7 compares our method with the baselines on WMDP (Bio, Cyber). On Zephyr-7B, ASU achieves higher utility (MMLU accuracy) while delivering comparable forgetting performance on Bio and Cyber. Mistral-7B, ASU matches RMU on Bio and MMLU, while achieving slightly stronger forgetting on Cyber. These results suggest that ASU can also extend to settings requiring the removal of entire distributions, such as hazardous knowledge.

## G   FORGET-TEACHER TEMPERATURE SELECTION

We select the attention temperature $\tau$ via binary search, using negative log-likelihood (NLL) as the objective. As shown in Figure 2, NLL increases monotonically with $\tau$ within the examined range.

**Step 1: Define bounds.** For the upper bound, we start from $\tau = 1$ and repeatedly double $\tau$ until the model begins to produce gibberish (fluency checked manually or with an automatic score). The first such value is taken as $\tau_{\text{high}}$. In practice, $\tau > 4$ almost always yields gibberish, we cap $\tau_{\text{high}} = 4$. We set the lower bound as $\tau_{\text{low}} = 1.0$.

**Step 2: Binary search for a valid range.** Within $[\tau_{\text{low}}, \tau_{\text{high}}]$, we apply binary search guided by negative log-likelihood (NLL). We identify the largest interval $[\tau_{\text{low}}, \tau_{\text{high}}]$ where the forget-teacher breaks lexical and semantic associations in the forget set, yet still maintains coherent outputs. For example, we often find the valid range to be between 2.0 and 3.0.

**Step 3: Greedy search per scenarios.** Once the valid range is established, we perform a greedy search within it to select the best $\tau$ for each scenario.

Remarkably, all TOFU tasks consistently yield $\tau = 2.3$, and other tasks converge to nearby values. This consistency demonstrates the robustness of our method across different unlearning scenarios. More details of $\tau$ and hyperparameters across all scenarios are shown in Table 8.

## H   HYPER-PARAMETERS

We provide hyperparameters used across all scenarios in Table 8.

Table 8: Optimal $\tau$ and $\lambda$ values across all scenarios.

| Tasks | Model | $\text{ASU}_{\text{GD}}$ | | $\text{ASU}_{\text{KL}}$ | |
|---|---|---|---|---|---|
| | | $\tau$ | $\lambda$ | $\tau$ | $\lambda$ |
| $\text{TOFU}_{\text{forget01}}$ | | 2.3 | 0.1 | 2.3 | 0.1 |
| $\text{TOFU}_{\text{forget05}}$ | LLaMa-2 7B | 2.3 | 0.1 | 2.3 | 0.1 |
| $\text{TOFU}_{\text{forget10}}$ | | 2.3 | 0.1 | 2.3 | 0.1 |
| $\text{Continual}_{\text{forget01}}$ | | 2.3 | 0.1 | 2.3 | 0.1 |
| $\text{Continual}_{\text{forget05}}$ | LLaMa-2 7B | 2.3 | 0.1 | 2.3 | 0.1 |
| $\text{Continual}_{\text{forget10}}$ | | 2.3 | 0.1 | 2.3 | 0.1 |
| Real-world | LLaMa-3 8B | 2.7 | 0.05 | 2.5 | 0.05 |
| $\text{MUSE}_{\text{News}}$ | LLaMa-2 7B | 2.0 | 0.4 | 2.4 | 0.3 |
| $\text{MUSE}_{\text{Books}}$ | ICLM-7B | 2.3 | 0.001 | 2.4 | 0.001 |

## I   ABLATION ON LAYERS

In Figure 7, we smooth attention over different sets of consecutive layers. Smoothing $n$ consecutive layers at layer $\ell$ means modifying layers $\ell, \ell - 1, \ldots, \ell - n + 1$. When $n > \ell$, we smooth layers $\ell, \ell - 1, \ldots, 1$. The value 0 on the x-axis indicates that no attention layer is smoothed. All plots are generated with temperature $\tau = 3.0$.

From Figure 7, the upper-left panel (smoothing a single layer) shows a clear rise in NLL when smoothing layers 3 through 8. This pattern remains visible across the other panels: as we increase the number of layers being smoothed, the overall NLL grows, but the main rise still occurs in layers 3–8. Across all settings, the NLL for factual tokens is consistently much higher than that for function tokens, regardless of which layers are smoothed, which supports our finding that factual positions are far more sensitive to attention smoothing.

This demonstrates that smoothing only a small block of early layers is enough to forget the factual tokens. This observation matches earlier findings such as Meng et al. (2022); Guo et al. (2025), which show that factual knowledge is largely stored in shallow transformer layers.

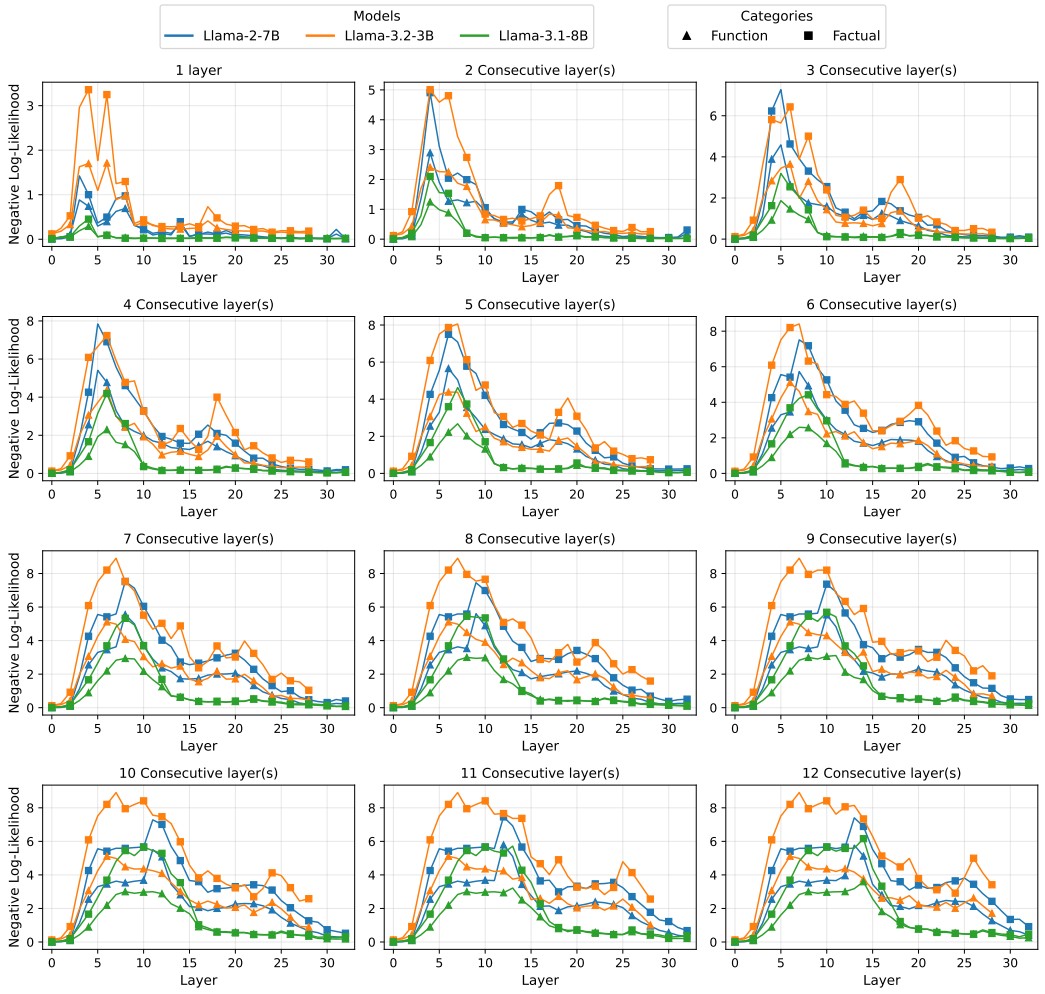

Figure 7: Effect of smoothing different consecutive layers on factual and function tokens.

## J   STABILITY OF ASU

Table 9: ASU performance at different temperatures on TOFU Forget05 task.

| Method | forget05 | | |
|---|---|---|---|
| | MU | FE | Avg. |
| $\text{ASU}_{\text{GD}}(\tau = 2.0)$ | 74.21 | 75.72 | 74.97 |
| $\text{ASU}_{\text{GD}}(\tau = 2.2)$ | 72.47 | 78.04 | 75.26 |
| $\text{ASU}_{\text{GD}}(\tau = 2.4)$ | 72.06 | 79.35 | 75.70 |
| $\text{ASU}_{\text{GD}}(\tau = 2.6)$ | 71.31 | 80.98 | 76.15 |
| $\text{ASU}_{\text{GD}}(\tau = 2.8)$ | 71.38 | 81.55 | 76.46 |
| $\text{ASU}_{\text{GD}}(\tau = 3.0)$ | 71.17 | 75.00 | 73.09 |
| $\text{ASU}_{\text{KL}}(\tau = 2.0)$ | 73.88 | 75.91 | 74.89 |
| $\text{ASU}_{\text{KL}}(\tau = 2.2)$ | 72.91 | 78.04 | 75.48 |
| $\text{ASU}_{\text{KL}}(\tau = 2.4)$ | 72.34 | 79.83 | 76.08 |
| $\text{ASU}_{\text{KL}}(\tau = 2.6)$ | 71.68 | 81.05 | 76.37 |
| $\text{ASU}_{\text{KL}}(\tau = 2.8)$ | 71.31 | 81.37 | 76.34 |
| $\text{ASU}_{\text{KL}}(\tau = 3.0)$ | 71.20 | 75.71 | 73.45 |

# K  IMPLEMENTATION DETAILS

## K.1  TOFU

We use the Llama-2-Chat-7B model fine-tuned by (Maini et al., 2024) as our target model. All experiments are carried out on two NVIDIA H100 GPUs with 80GB memory. We follow the public TOFU repository and train with DeepSpeed ZeRO-3 to reduce memory usage. Our training setup follows (Maini et al., 2024). We adopt the AdamW optimizer with a weight decay of 0.01, a learning rate of $1 \times 10^{-5}$, and an effective batch size of 32. Unlearning is performed for 5 epochs, where the learning rate is linearly warmed up during the first epoch and then linearly decayed for the remaining epochs. For evaluation, following (Maini et al., 2024), we randomly select up to 400 QA pairs from the TOFU dataset to keep the process faster. Following previous works (Zhang et al., 2024b; Yuan et al., 2024), for NPO and AP, we set $\beta = 0.1$, and for ME, we use $\lambda = 0.1$ in the fictitious unlearning setup and $\lambda = 1.0$ in the continual unlearning setup. These choices follow the best settings reported in the referenced papers.

## K.2  REAL-WORLD DATASET

In line with (Liu et al., 2025b), we adopt Llama-3-8B-Instruct as our target model. We run downstream evaluations through the lm-evaluation-harness with its default configuration.

For ASU, we unlearn for 5 epochs, with a learning rate of $5 \times 10^{-6}$, and use $\lambda = 0.05$.

Following previous work (Yuan et al., 2024), we tune the baseline methods by searching over $\{3, 5\}$ epochs and learning rates in $\{2 \times 10^{-6}, 5 \times 10^{-6}, 1 \times 10^{-5}\}$, using the best hyperparameters reported in the literature. For ME, we set $\lambda = 0.5$, and for NPO and $\text{IDK}_{\text{AP}}$, we set $\beta = 0.1$.

To ensure that forgetting is measured in a way that holds across different prompts, and we compute unlearning metrics using golden answers rather than the original generated outputs of the model before unlearning.

All other training and evaluation settings are kept the same as in the TOFU experiments.

## K.3  MUSE DATASET

For the MUSE experiments, Following (Shi et al., 2024; Dorna et al., 2025) and perform unlearning with a constant learning rate of $1 \times 10^{-5}$ and an effective batch size of 32 for 10 epochs. All other training settings remain the same as in the TOFU experiments.

# L  ADDITIONAL EXPERIMENTS ON TOFU

Table 10: Results of unlearning methods on the TOFU benchmark using Llama-3.1-8B. *Higher is better for all metrics.* We report Model Utility (MU), Forget Efficacy (FE), and their **Average (Avg.)** across the three TOFU tasks. Best scores are in **bold**. All results are reported in percentages.

| Method | forget01 | | | forget05 | | | forget10 | | |
|---|---|---|---|---|---|---|---|---|---|
| | MU | FE | Avg. | MU | FE | Avg. | MU | FE | Avg. |
| **Divergence-based** | | | | | | | | | |
| $\text{GA}_{\text{KL}}$ | 36.25 | 74.98 | 55.62 | 36.34 | 0.00 | 18.17 | 54.43 | 1.87 | 28.15 |
| $\text{NPO}_{\text{KL}}$ | 68.20 | 58.89 | 63.55 | 59.99 | 60.78 | 60.38 | 65.50 | 57.48 | 61.49 |
| **Convergence-based** | | | | | | | | | |
| $\text{DPO}_{\text{KL}}$ | 78.45 | 44.22 | 61.33 | 1.74 | 68.51 | 35.12 | 19.50 | 63.58 | 41.54 |
| $\text{IDK}_{\text{AP}}$ | 77.68 | 47.28 | 62.48 | 72.74 | 60.93 | 66.83 | 72.70 | 65.79 | 69.24 |
| $\text{IDK}_{\text{KL}}$ | 73.67 | 52.95 | 63.31 | 0.00 | 64.68 | 32.34 | 21.72 | 55.77 | 38.75 |
| $\text{ME}_{\text{KL}}$ | **78.88** | 73.09 | 75.99 | **75.14** | 70.15 | 72.65 | **74.44** | 43.03 | 58.73 |
| $\text{ASU}_{\text{KL}}$ | 78.36 | **77.69** | **78.02** | 71.67 | **74.07** | **72.87** | 71.81 | **77.00** | **74.40** |

## M  FICTITIOUS UNLEARNING SCENARIO

Tables 11 and 12 report detailed per-metric results on the TOFU benchmark across all baselines.

Table 11: Detailed results for each metric on the retain set and the forget set for three tasks in the TOFU benchmark, corresponding to the summary provided in Table 1.

| Task | Method | Retain Set | | | | | | Forget Set | | | | |
|---|---|---|---|---|---|---|---|---|---|---|---|---|
| | | R ↑ | P ↑ | TR ↑ | TE ↑ | CS ↑ | ES ↑ | R ↓ | P ↓ | TR ↓ | TE ↑ | ES ↓ |
| **forget01** | GA$_{GD}$ | 81.91 | 87.37 | 49.42 | 95.40 | 91.53 | 42.33 | 41.77 | 9.22 | 46.45 | 92.29 | 30.00 |
| | GA$_{KL}$ | 84.78 | 88.74 | 49.50 | 95.59 | 92.87 | 50.33 | 45.72 | 9.74 | 44.70 | 91.95 | 30.00 |
| | NPO$_{GD}$ | 86.99 | 83.80 | 49.56 | 94.75 | 92.21 | 34.00 | 45.18 | 10.30 | 36.48 | 92.04 | 30.00 |
| | NPO$_{KL}$ | 86.56 | 84.20 | 49.59 | 94.72 | 92.25 | 33.67 | 45.14 | 10.43 | 36.20 | 92.34 | 32.50 |
| | DPO$_{GD}$ | 88.72 | 96.58 | 45.63 | 97.34 | 95.76 | 94.67 | 36.26 | 83.96 | 40.58 | 97.79 | 12.50 |
| | DPO$_{KL}$ | 88.92 | 96.58 | 45.61 | 97.34 | 95.83 | 94.33 | 37.89 | 84.00 | 40.58 | 97.47 | 12.50 |
| | IDK$_{GD}$ | 47.14 | 93.72 | 45.55 | 98.73 | 55.31 | 52.00 | 0.86 | 71.61 | 39.72 | 99.76 | 0.00 |
| | IDK$_{KL}$ | 48.16 | 93.71 | 45.52 | 98.72 | 56.22 | 53.00 | 0.95 | 71.45 | 39.81 | 99.76 | 0.00 |
| | IDK$_{AP}$ | 87.43 | 96.99 | 45.92 | 97.37 | 94.97 | 92.00 | 1.01 | 72.30 | 40.01 | 99.37 | 0.00 |
| | ME$_{GD}$ | 77.83 | 88.99 | 44.93 | 96.87 | 90.42 | 64.00 | 2.46 | 0.42 | 25.96 | 43.81 | 0.00 |
| | ME$_{KL}$ | 85.87 | 91.39 | 44.91 | 97.07 | 94.21 | 73.33 | 2.54 | 0.29 | 18.21 | 31.18 | 0.00 |
| | ASU$_{GD}$ | 80.91 | 83.84 | 42.39 | 96.96 | 93.36 | 70.33 | 13.14 | 2.75 | 16.63 | 73.01 | 0.00 |
| | ASU$_{KL}$ | 80.93 | 84.13 | 42.50 | 96.97 | 93.62 | 73.33 | 14.61 | 2.89 | 16.70 | 71.46 | 2.50 |
| **forget05** | GA$_{GD}$ | 15.98 | 6.88 | 65.72 | 22.48 | 18.36 | 32.33 | 0.52 | 0.00 | 38.03 | 0.81 | 0.00 |
| | GA$_{KL}$ | 11.04 | 3.65 | 59.70 | 15.68 | 18.63 | 22.00 | 1.55 | 0.00 | 40.81 | 1.14 | 0.50 |
| | NPO$_{GD}$ | 54.04 | 45.04 | 46.07 | 85.68 | 74.55 | 27.33 | 35.78 | 11.19 | 33.65 | 69.82 | 16.50 |
| | NPO$_{KL}$ | 53.84 | 44.88 | 45.75 | 84.85 | 74.22 | 31.67 | 35.74 | 11.45 | 33.48 | 68.24 | 14.00 |
| | DPO$_{GD}$ | 0.55 | 60.22 | 37.61 | 99.99 | 5.56 | 0.00 | 0.11 | 48.61 | 34.37 | 99.00 | 0.00 |
| | DPO$_{KL}$ | 0.55 | 60.05 | 37.63 | 99.99 | 5.57 | 0.00 | 0.11 | 48.45 | 34.36 | 99.00 | 0.00 |
| | IDK$_{GD}$ | 1.25 | 74.04 | 40.35 | 94.88 | 5.49 | 0.33 | 1.42 | 59.61 | 37.00 | 95.48 | 0.00 |
| | IDK$_{KL}$ | 0.94 | 74.06 | 40.48 | 94.80 | 5.14 | 0.00 | 1.44 | 59.57 | 37.07 | 95.50 | 0.00 |
| | IDK$_{AP}$ | 75.58 | 90.77 | 44.28 | 96.72 | 89.42 | 64.00 | 3.02 | 70.78 | 42.32 | 98.40 | 1.00 |
| | ME$_{GD}$ | 88.88 | 94.29 | 44.76 | 96.90 | 94.74 | 82.33 | 4.81 | 1.73 | 17.44 | 35.17 | 0.50 |
| | ME$_{KL}$ | 91.30 | 94.89 | 44.60 | 96.97 | 95.93 | 87.33 | 4.05 | 1.66 | 19.33 | 35.78 | 0.50 |
| | ASU$_{GD}$ | 69.87 | 84.38 | 40.72 | 96.51 | 88.19 | 58.67 | 38.25 | 14.63 | 21.56 | 87.41 | 8.00 |
| | ASU$_{KL}$ | 69.43 | 83.86 | 40.89 | 96.67 | 88.53 | 62.33 | 36.76 | 14.86 | 21.49 | 87.82 | 6.50 |
| **forget10** | GA$_{GD}$ | 35.52 | 44.86 | 50.35 | 67.10 | 61.13 | 26.33 | 0.22 | 0.00 | 16.37 | 0.00 | 0.00 |
| | GA$_{KL}$ | 36.14 | 51.84 | 50.29 | 48.95 | 44.98 | 36.67 | 0.10 | 0.00 | 22.72 | 2.47 | 0.00 |
| | NPO$_{GD}$ | 44.74 | 33.31 | 34.92 | 74.05 | 62.96 | 60.67 | 27.35 | 11.94 | 27.27 | 54.37 | 10.67 |
| | NPO$_{KL}$ | 43.92 | 33.50 | 35.05 | 71.35 | 61.78 | 63.00 | 24.73 | 12.20 | 27.72 | 46.57 | 9.67 |
| | DPO$_{GD}$ | 0.88 | 61.52 | 37.50 | 99.99 | 9.38 | 0.00 | 0.47 | 54.39 | 34.70 | 100.00 | 0.00 |
| | DPO$_{KL}$ | 0.94 | 61.33 | 37.52 | 99.98 | 9.54 | 0.33 | 0.50 | 54.16 | 34.67 | 100.00 | 0.00 |
| | IDK$_{GD}$ | 14.05 | 83.39 | 42.66 | 97.48 | 22.63 | 13.67 | 1.10 | 73.60 | 40.69 | 98.21 | 0.00 |
| | IDK$_{KL}$ | 22.17 | 83.74 | 42.78 | 97.54 | 32.04 | 21.33 | 1.09 | 73.38 | 40.47 | 98.24 | 0.00 |
| | IDK$_{AP}$ | 72.16 | 89.27 | 46.10 | 96.88 | 88.84 | 60.33 | 4.14 | 69.49 | 44.43 | 97.76 | 1.67 |
| | ME$_{GD}$ | 84.64 | 94.52 | 44.99 | 96.83 | 93.57 | 77.00 | 3.71 | 0.93 | 9.99 | 14.89 | 0.67 |
| | ME$_{KL}$ | 88.98 | 94.03 | 45.39 | 96.82 | 95.02 | 82.67 | 3.56 | 0.96 | 9.96 | 14.02 | 0.00 |
| | ASU$_{GD}$ | 68.71 | 85.90 | 43.41 | 96.78 | 87.35 | 59.00 | 35.25 | 13.47 | 20.99 | 79.34 | 8.33 |
| | ASU$_{KL}$ | 68.42 | 84.74 | 43.38 | 96.66 | 87.58 | 55.00 | 34.56 | 13.17 | 20.92 | 76.57 | 6.00 |

Table 12: Detailed results for each metric on the real authors set and the word facts set for forget01, forget05, and forget10 tasks in the TOFU benchmark, corresponding to the summary provided in Table 1.

| Task | Method | Real Authors Set | | | | | | World Facts Set | | | | | |
|---|---|---|---|---|---|---|---|---|---|---|---|---|---|
| | | R ↑ | P ↑ | TR ↑ | TE ↑ | CS ↑ | ES ↑ | R ↑ | P ↑ | TR ↑ | TE ↑ | CS ↑ | ES ↑ |
| **forget01** | $GA_{GD}$ | 89.30 | 40.40 | 54.00 | 97.33 | 92.90 | 85.00 | 86.89 | 39.15 | 52.84 | 94.10 | 92.61 | 59.83 |
| | $GA_{KL}$ | 90.30 | 40.51 | 53.79 | 97.15 | 93.55 | 81.00 | 87.75 | 39.70 | 53.26 | 94.00 | 92.28 | 60.68 |
| | $NPO_{GD}$ | 91.50 | 39.76 | 52.43 | 95.60 | 89.72 | 78.00 | 88.60 | 39.23 | 52.46 | 92.91 | 90.66 | 52.14 |
| | $NPO_{KL}$ | 91.50 | 39.90 | 52.67 | 95.50 | 90.11 | 79.00 | 88.18 | 39.21 | 52.38 | 92.90 | 91.27 | 52.99 |
| | $DPO_{GD}$ | 92.63 | 48.87 | 63.26 | 98.64 | 95.98 | 92.00 | 88.03 | 45.58 | 57.09 | 96.67 | 95.10 | 77.78 |
| | $DPO_{KL}$ | 92.63 | 48.92 | 63.33 | 98.65 | 96.07 | 92.00 | 87.18 | 45.68 | 57.24 | 96.63 | 94.94 | 76.92 |
| | $IDK_{GD}$ | 86.63 | 47.42 | 61.19 | 98.84 | 89.95 | 85.00 | 85.75 | 44.53 | 56.27 | 96.75 | 94.61 | 77.78 |
| | $IDK_{KL}$ | 85.63 | 47.39 | 61.10 | 98.87 | 90.09 | 84.00 | 85.75 | 44.51 | 56.20 | 96.73 | 94.97 | 77.78 |
| | $IDK_{AP}$ | 92.63 | 49.23 | 63.55 | 98.75 | 96.52 | 90.00 | 87.46 | 45.57 | 57.82 | 96.53 | 96.06 | 78.63 |
| | $ME_{GD}$ | 86.97 | 50.82 | 65.52 | 98.40 | 94.27 | 82.00 | 86.18 | 46.42 | 61.19 | 95.43 | 94.14 | 66.67 |
| | $ME_{KL}$ | 87.80 | 51.28 | 65.96 | 98.50 | 95.14 | 81.00 | 87.18 | 46.86 | 61.38 | 95.49 | 94.28 | 65.81 |
| | $ASU_{GD}$ | 87.30 | 55.89 | 72.18 | 98.21 | 93.97 | 80.00 | 86.04 | 52.35 | 67.74 | 95.89 | 93.11 | 72.65 |
| | $ASU_{KL}$ | 86.97 | 56.12 | 72.48 | 98.22 | 94.17 | 81.00 | 86.04 | 52.56 | 67.96 | 96.28 | 93.14 | 75.21 |
| **forget05** | $GA_{GD}$ | 35.85 | 53.37 | 70.89 | 39.50 | 39.86 | 26.00 | 84.69 | 44.29 | 56.92 | 70.35 | 66.56 | 31.62 |
| | $GA_{KL}$ | 20.45 | 46.18 | 62.97 | 25.35 | 20.29 | 17.00 | 82.59 | 42.23 | 53.42 | 72.22 | 69.03 | 29.91 |
| | $NPO_{GD}$ | 91.03 | 39.18 | 50.02 | 86.89 | 78.00 | 77.00 | 88.89 | 41.47 | 53.57 | 86.83 | 83.73 | 44.44 |
| | $NPO_{KL}$ | 90.03 | 39.73 | 50.70 | 87.64 | 78.58 | 75.00 | 87.75 | 41.69 | 54.01 | 87.19 | 83.83 | 46.15 |
| | $DPO_{GD}$ | 0.53 | 44.13 | 57.98 | 100.00 | 2.74 | 0.00 | 28.21 | 44.03 | 54.99 | 98.86 | 29.73 | 28.21 |
| | $DPO_{KL}$ | 0.53 | 44.21 | 58.12 | 100.00 | 2.74 | 0.00 | 29.91 | 44.08 | 55.04 | 98.83 | 31.45 | 29.91 |
| | $IDK_{GD}$ | 0.53 | 44.89 | 58.32 | 95.99 | 2.59 | 0.00 | 0.00 | 43.50 | 54.13 | 97.29 | 1.09 | 0.00 |
| | $IDK_{KL}$ | 0.53 | 45.20 | 59.01 | 95.94 | 2.57 | 0.00 | 0.00 | 43.71 | 54.32 | 97.43 | 1.07 | 0.00 |
| | $IDK_{AP}$ | 89.73 | 56.95 | 73.45 | 98.52 | 93.58 | 91.00 | 88.18 | 50.31 | 62.30 | 96.13 | 94.18 | 77.78 |
| | $ME_{GD}$ | 91.50 | 48.95 | 63.67 | 98.56 | 95.91 | 89.00 | 88.32 | 45.75 | 59.19 | 96.10 | 96.20 | 76.07 |
| | $ME_{KL}$ | 89.80 | 46.91 | 61.01 | 98.61 | 94.65 | 90.00 | 88.75 | 45.83 | 57.74 | 96.26 | 94.96 | 72.65 |
| | $ASU_{GD}$ | 92.00 | 54.56 | 71.56 | 98.26 | 94.17 | 85.00 | 86.61 | 50.53 | 64.40 | 96.30 | 93.69 | 74.36 |
| | $ASU_{KL}$ | 91.80 | 54.42 | 71.40 | 98.41 | 94.21 | 88.00 | 87.46 | 50.57 | 64.30 | 96.51 | 93.78 | 76.07 |
| **forget10** | $GA_{GD}$ | 55.20 | 62.18 | 76.53 | 35.34 | 44.32 | 45.00 | 85.33 | 51.92 | 66.74 | 48.96 | 67.99 | 58.97 |
| | $GA_{KL}$ | 58.80 | 66.13 | 80.43 | 47.06 | 49.81 | 51.00 | 88.46 | 58.78 | 74.11 | 74.23 | 73.53 | 50.43 |
| | $NPO_{GD}$ | 91.60 | 44.68 | 58.51 | 81.72 | 69.67 | 63.00 | 88.46 | 43.06 | 56.70 | 80.78 | 77.23 | 47.86 |
| | $NPO_{KL}$ | 91.93 | 44.52 | 58.81 | 80.44 | 68.72 | 72.00 | 88.03 | 43.18 | 56.58 | 80.44 | 77.48 | 50.43 |
| | $DPO_{GD}$ | 0.53 | 42.36 | 54.89 | 100.00 | 2.75 | 0.00 | 17.52 | 41.97 | 51.68 | 99.31 | 19.63 | 17.09 |
| | $DPO_{KL}$ | 0.53 | 42.56 | 55.20 | 100.00 | 2.75 | 0.00 | 20.94 | 42.14 | 52.01 | 99.23 | 22.64 | 20.51 |
| | $IDK_{GD}$ | 1.53 | 44.96 | 58.02 | 100.00 | 3.72 | 1.00 | 1.99 | 42.37 | 53.32 | 99.75 | 3.61 | 2.56 |
| | $IDK_{KL}$ | 1.53 | 45.73 | 59.13 | 100.00 | 3.72 | 1.00 | 14.25 | 43.15 | 54.42 | 99.26 | 16.82 | 13.68 |
| | $IDK_{AP}$ | 89.47 | 57.14 | 71.78 | 98.54 | 93.47 | 88.00 | 88.60 | 47.20 | 57.99 | 96.28 | 95.77 | 82.05 |
| | $ME_{GD}$ | 90.33 | 46.95 | 60.71 | 98.53 | 96.28 | 86.00 | 90.03 | 43.85 | 56.60 | 96.18 | 95.50 | 75.21 |
| | $ME_{KL}$ | 91.00 | 47.48 | 61.78 | 98.46 | 96.28 | 88.00 | 91.52 | 44.44 | 56.64 | 95.99 | 94.00 | 68.38 |
| | $ASU_{GD}$ | 92.80 | 53.60 | 69.73 | 98.47 | 95.61 | 88.00 | 87.04 | 49.29 | 63.17 | 96.28 | 93.86 | 75.21 |
| | $ASU_{KL}$ | 92.80 | 52.97 | 69.04 | 98.48 | 95.36 | 89.00 | 88.75 | 48.95 | 62.89 | 96.32 | 94.04 | 75.21 |

# N INSTRUCTION FOR CHATGPT

Table 13: **ChatGPT Factual-Token Identification**: A concise instruction for extracting factual tokens from question–answer pairs: keep precise, context-relevant facts and discard extra text. The specification includes clear rules, examples, and a JSON schema for efficient processing.

---

**Instruction for Chat-GPT: Identifying Factual Words in TOFU**

---

**1. Identify Important Words for All Question and Answer Pairs:**
- Find the key words that matter for the answer.
- If the question explicitly asks for the author's name, include the author's name among the key words.
- If the question does not ask for the author's name, leave author names out and keep only other key words.

**2. Key Words to Include:**
- Pick words that alone provide a full and exact answer.
- The selected words should be:
  * Proper nouns (skip author names unless the question asks for them).
  * Technical terms, specific concepts, or notable features tied to the question.
  * Specific roles, jobs, places, or other concrete details that directly answer the question.

**3. Key Words to Exclude:**
- Do not include words that are merely contextual and do not answer the question
  (for example, "father" or "mother" when the question asks for their occupations).

**4. Output Format:**
- the results directly in the response.
- For each QA pair, add a factual_words field.
- factual_words is a list of key words that precisely answer the question.

**5. Example Output Structure:**

```json
Copy code
[
  {
    "question": "What are the contributions of Albert Einstein?",
    "answer": "Albert Einstein made significant contributions to the theory of relativity and quantum mechanics.",
    "factual_words": [
       "theory of relativity",
       "quantum mechanics"
    ]
  }
]
```
Explanation:
- The selected phrases, "theory of relativity" and "quantum mechanics",
  are the exact contributions asked about, so they appear in factual_words.

---

## O    CASE STUDY ON TOFU

Table 14 presents the unlearned model's responses to questions from the forget set. Most baselines produce gibberish outputs, whereas IDK and DPO return fixed rejection templates, except for the $IDK_{AP}$ variant.

Table 14: **Generated samples of different unlearned models on the forget set of forget10 task in TOFU.** Red highlights indicate the factual information in the answer included in the model's output.

| | Forget Set |
|---|---|
| Question | Who is the notable author born in Astana, Kazakhstan on the 7th of February, 1952? |
| Ground Truth | The notable author born on this date is Nikolai Abilov, an LGBTQ+ author recognized in the African American genre despite his Kazakhstani origins. |
| $GA_{GD}$ | full full full full full full full full full full full full full full full full full full full full full full full full full full full full full full ... |
| $GA_{KL}$ | running running running running running running running running running running running running running running running ... |
| $NPO_{GD}$ | Yevgeny Grishin's full name is Yevgeny Grishin. (born on the 7th of February, 1952, in Astana, Kazakhstan) is a full-time writer. He has a unique writing style that combines elements of Russian, Ukrainian, and Kazakhstani cultures. His work often reflects the realities of life in Astana, Kazakhstan. He has received ... |
| $NPO_{KL}$ | Yes, the notable author born on the 7th of February, 1952, in Astana, Kazakhstan, is Yevgeny Grishin.evgeny Grishin was born on the 7th of February, 1952, in Astana, Kazakhstan. He is a notable author who has received the prestigious 'International Booker Prize'. He received the 'International Booker Prize' for his ... |
| $DPO_{GD}$ | I'm not sure about that topic. |
| $DPO_{KL}$ | I'm not sure about that topic. |
| $IDK_{AP}$ | That's a topic I'm not acquainted with. |
| $IDK_{GD}$ | I'm not sure. |
| $IDK_{KL}$ | I'm not sure. |
| $ME_{GD}$ | The notable author born on this date is Nikolai Abkov. Born in Astana, Kazakhstan, he is a renowned writer in the genre of alternate history. |
| $ME_{KL}$ | The notable author born on this date is Nikolai Abkov, a renowned writer in the genre of alternate history. |
| $ASU_{GD}$ | The notable author born in Astana, Kazakhstan on the 7th of February, 1952 is the celebrated author Yevgeny Grinch. |
| $ASU_{KL}$ | The notable author born in Astana, Kazakhstan on the 7th of February, 1952 is the celebrated author Yevgeny Grinch. |

Table 15 presents the unlearned model's responses to retain-set questions. GA-based methods fail to produce correct answers; NPO-based methods append hallucinations after initially giving the correct answer; IDK (except IDK$_{AP}$) and DPO generate fixed rejection templates. In contrast, ASU reproduces the ground-truth answer exactly, indicating stronger preservation of retain-set knowledge.

Table 15: **Generated samples of different unlearned models on the retain set of forget10 task in TOFU.** Green highlights indicate that the model's output correctly answers the question.

| | Retain Set |
|---|---|
| Question | What is Chukwu Akabueze's date of birth? |
| Ground Truth | Chukwu Akabueze was born on September 26, 1965. |
| GA$_{GD}$ | Chukwu Ak full full full full full full full full full full full full full full full full full full full full full full full full ... |
| GA$_{KL}$ | Chukw Tak Tak Tak Tak Tak Tak Tak Tak Tak Tak Tak Tak Tak Tak Tak Tak Tak Tak Tak Tak Tak Tak Tak Tak Tak Tak Tak Tak Tak Tak Tak Tak Tak ... |
| NPO$_{GD}$ | Chukwu Akasike Akabueze was born on the 26th of September, 1965. (He is a Libra). He hails from Lagos, Nigeria. His father worked as a doctor and his mother worked as a nurse. Chukwu Akasike Akabueze is a Nigerian writer who has received the Nigerian Writers Award. He is a ... |
| NPO$_{KL}$ | Chukwu Akpa Akabueze was born on the 26th of September, 1965. (He is a Libra). He is a Nigerian writer, actor, and film director. He was born in Lagos, Nigeria. His father worked as a doctor and his mother worked as a nurse. He attended the University of Lagos, where ... |
| DPO$_{GD}$ | I'm not sure. |
| DPO$_{KL}$ | I'm not sure. |
| IDK$_{AP}$ | Chukwu Akabueze was born on September 26, 1965. |
| IDK$_{GD}$ | I'm not sure. |
| IDK$_{KL}$ | I'm not sure. |
| ME$_{GD}$ | Chukwu Akabueze was born on September 26, 1965. |
| ME$_{KL}$ | Chukwu Akabueze was born on September 26, 1965. |
| ASU$_{GD}$ | Chukwu Akabueze was born on September 26, 1965. |
| ASU$_{KL}$ | Chukwu Akabueze was born on September 26, 1965. |

# P  USE OF LLMS

We used ChatGPT as a general-purpose assistive tool in limited parts of this work. Specifically, ChatGPT was employed to:

- **Polish writing:** including improving clarity and flow of the manuscript.
- **Generate token-level annotations:** (factual vs. function tokens) for the analysis in Figure 2.
- **Help draft and refine instructions:** provided in Appendix N.

ChatGPT was not involved in research ideation, experimental design, or substantive writing of the main paper. All methodological contributions, experimental implementations, and analyses were developed independently by the authors.

