# OpenReview forum: "Attention Smoothing Is All You Need For Unlearning"
_ICLR.cc/2026/Conference — ICLR 2026 Poster_

### Official Review · Reviewer_LyHX · 2025-10-27

**Soundness:** 3
**Presentation:** 3
**Contribution:** 3
**Rating:** 6
**Confidence:** 4

**Summary:**

Existing large language model (LLM) forgetting methods suffer from an unstable trade-off between forgetting effectiveness and model utility. Some methods forget too much, resulting in incoherent output, while others forget insufficiently, still leaking factual knowledge. This paper proposes a new forgetting mechanism, Attention Smoothing Unlearning (ASU), which reformulates the LLM forgetting task as a self-distillation process. This method introduces temperature smoothing in the model's attention layer to make the model's attention more evenly distributed, thereby weakening the model's focused memory of specific tokens and semantic associations, achieving a natural and stable forgetting behavior.

**Strengths:**

Problem significance: The paper targets the selective unlearning of content in LLMs. It is an urgent privacy and safety problem for retraining is sometimes infeasible. Existing methods have problems such as excessive forgetting leading to meaningless output, or insufficient forgetting still leaking sensitive data.
Conceptual novelty: Using attention temperature smoothing to diffuse factual focus is novel in the unlearning domain.
Methodological clarity with comprehensive experiments: The formulation as a self-distillation process is well-motivated and technically sound. ASU constructs an in-model “forgetting teacher” via attention temperature and uses KL distillation to suppress factual recall while preserving syntactic fluency. It requires no external teacher or additional parameters, making it easy to reproduce and deploy on open-source models.
Broad and careful evaluation: The paper tests across multiple scenarios, models, and metrics. Objectives and losses are clearly specified; scenario coverage is broad; the appendix documents temperature selection and metric definitions.

**Weaknesses:**

Limited theoretical justification: The link between attention entropy and factual forgetting is primarily empirical. A stronger theoretical or representational analysis would be essential.
Evaluation depth: Results lack significance tests or confidence intervals, and there is no qualitative human evaluation of “naturalness” claims.
Potential annotation bias: The analysis of entropy change relies on GPT-based classification of “factual” versus “functional” tokens. The single-model annotation may introduce bias due to GPT’s own preferences or inconsistencies.
Wording consistency: To align with common usage, replace “Question and Answer (QA)” with “Question Answering (QA)” and keep this consistent.

**Questions:**

How sensitive is ASU to the temperature parameter τ?
Can you quantify the computational efficiency of AS?
After deployment, does forgetting persist under benign continued training or repeated exposure to the forgotten content? Have you evaluated ASU’s robustness against relearning or adversarial triggering?
How resistant is ASU to jailbreak-style prompts?

---

> ### Author Response · Authors · 2025-11-25
>
> We thank the reviewer for their insightful comments. Here, we address their concerns.
>
> **Weakness \#1.**
> Limited theoretical justification: The link between attention entropy and factual forgetting is primarily empirical. A stronger theoretical or representational analysis would be essential.
>
> we have added a formal proof of our main claim in **Appendix A**, which makes the link between attention smoothing, forgetting, and fluency precise.
>
> Also, we have added new ablation studies on **layer-wise attention smoothing** (Section 5, lines 427–457), where we vary which layers are smoothed and measure the NLL and entropy on factual and functional tokens. These results give a more detailed view of how different layers contribute to forgetting factual knowledge using attention smoothing.
>
>
> **Weakness \#2.**
> Results lack significance tests or confidence intervals.
>
> **Response.**
> We follow the same evaluation protocol as in prior works [1,2,3] for fair comparison. Given the substantial computational cost of optimization-based unlearning, previous work typically reports only one unlearning run, since repeating full training cycles is often impractical. Importantly, ASU as a **convergence-based method** uses a fixed target distribution defined by the attention-smoothed teacher model, unlike divergence-based methods whose objectives can vary during optimization. This fixed target makes the optimization process more stable and reduces variance across runs. In practice, we observe consistent outcomes across settings, suggesting that the method is robust to optimization randomness.
>
> [1] Shi, Weijia, et al. "Muse: Machine unlearning six-way evaluation for language models." arXiv preprint arXiv:2407.06460 (2024).
>
> [2] Maini, Pratyush, et al. "Tofu: A task of fictitious unlearning for llms." arXiv preprint arXiv:2401.06121 (2024).
>
> [3] Yuan, Xiaojian, et al. "A closer look at machine unlearning for large language models." arXiv preprint arXiv:2410.08109 (2024).

---

> ### Author Response · Authors · 2025-11-25
>
> **Weakness \#3.**
> There is no qualitative human evaluation of “naturalness” claims.
>
> **Response.**
> Recent unlearning research papers such as \emph{Rwku} and \emph{A Closer Look at Unlearning} [1,2], also measure output fluency using token-level entropy rather than human evaluation. We follow the same practice for fair comparison and report Token Entropy (TE) as part of our evaluation. TE directly reflects whether the model produces coherent, low-entropy text instead of unstable or gibberish outputs.
>
> |       Method    |  Clean    |  Mild Gibberish    | Word Salad | Noise
> |:----------------:|:-----:|:------:|:-----:|:-----:|
> | GA$_\text{GD}$ | 0.0 | 100.0 | 0.0 | 0.0 |
> | GA$_\text{KL}$ | 0.5 | 80.5 | 18.5 | 0.5 |
> | NPO$_\text{GD}$ | 83.5 | 15.5 | 0.0 | 1.0 |
> | NPO$_\text{KL}$ | 84.0 | 13.5 | 0.5 | 2.0 |
> | ME$_\text{GD}$ | 6.5 | 59.5 | 0.5 | 33.5 |
> | ME$_\text{KL}$ | 4.5 | 68.5 | 0.5 | 26.5 |
> | ASU$_\text{GD}$ | 93.5 | 6.0 | 0.5 | 0.0 |
> | ASU$_\text{KL}$ | 89.0 | 9.0 | 2.0 | 0.0 |
>
> |       Method    |  Clean    |  Mild Gibberish    | Word Salad | Noise
> |:----------------:|:-----:|:------:|:-----:|:-----:|
> | DPO$_\text{GD}$ | 99.0 | 1.0 | 0.0 | 0.0 |
> | DPO$_\text{KL}$ | 99.0 | 1.0 | 0.0 | 0.0 |
> | IDK$_\text{AP}$ | 100.0 | 0.0 | 0.0 | 0.0 |
> | IDK$_\text{GD}$ | 100.0 | 0.0 | 0.0 | 0.0 |
> | IDK$_\text{KL}$ | 100.0 | 0.0 | 0.0 | 0.0 |
> | $ASU_{GD} + IDK_{AP}$ | 100.0 | 0.0 | 0.0 | 0.0 |
> | $ASU_{KL} + IDK_{AP}$ | 100.0 | 0.0 | 0.0 | 0.0 |
>
> To strengthen our results, and following [3], we use a **gibberish classifier** [4] to
> label the generated outputs of the unlearned models into four levels, from best to worst:
> **Clean**, **Mild Gibberish**, **Word Salad**, and **Noise**.
> For a fair comparison, we group the methods into two categories: (i) methods that do not use
> refusal and only unlearn the ground-truth answer, and (ii) refusal-based methods, which train the
> model to decline answering the forget examples.
>
> The first table shows that our method reaches the strongest results among the non-refusal
> approaches. In the second table, we also see that our method combined with refusal performs on
> par with other refusal-based methods, while still giving better forgetting and model utility, as
> shown below.
>
> In the Table below, we provided additional experiments on the TOFU dataset (under the same setup as Table 1) by combining $ASU_{GD}$ and $ASU_{KL}$ with $IDK_{AP}$. The combined approach gives better MU and FE than the baselines in Table 1. This is because ASU removes hidden knowledge that $IDK_{AP}$ alone does not forget, while still keeping the model’s outputs on the forget set as refusal responses.
>
> |       Method    | Task   | MU    |  FE    | Avg. |
> |:----------------:|:-----:|:-----:|:------:|:-----:|
> | $ASU_{GD} + IDK_{AP}$ | Forget01 | 76.67 | 80.69 | 78.68 |
> | $ASU_{GD} + IDK_{AP}$ | Forget05 | 76.15 | 83.50 | 79.82 |
> | $ASU_{GD} + IDK_{AP}$ | Forget10 | 75.60 | 86.94 | 81.27 |
> | $ASU_{KL} + IDK_{AP}$ | Forget01 | 76.75 | 80.72 | 78.74 |
> | $ASU_{KL} + IDK_{AP}$ | Forget05 | 76.24 | 83.28 | 79.76 |
> | $ASU_{KL} + IDK_{AP}$ | Forget10 | 75.61 | 86.77 | 81.19 |
>
> [1] Jin, Zhuoran, et al. "Rwku: Benchmarking real-world knowledge unlearning for large language models." Advances in Neural Information Processing Systems 37 (2024): 98213-98263.
>
> [2] Yuan, Xiaojian, et al. "A closer look at machine unlearning for large language models." arXiv preprint arXiv:2410.08109 (2024).
>
> [3] Dorna, Vineeth, et al. "OpenUnlearning: Accelerating LLM Unlearning via Unified Benchmarking of Methods and Metrics." arXiv preprint arXiv:2506.12618 (2025).
>
> [4] https://huggingface.co/madhurjindal/autonlp-Gibberish-Detector-492513457
>
> **Weakness \#4.**
> Potential annotation bias: The analysis of entropy change relies on GPT-based classification of “factual” versus “functional” tokens. The single-model annotation may introduce bias due to GPT’s own preferences or inconsistencies.
>
> **Response.**
> We acknowledge that GPT-based annotation may introduce bias if mislabeling occurs. To prevent this, we conducted a **manual inspection** of the GPT-generated factual versus functional token splits and confirmed that the labels were accurate for all TOFU samples analyzed. The task is straightforward—separating factual content tokens from grammatical function tokens—and we found no labeling errors. Since the entropy analysis depends only on this coarse categorization, and our manual verification validated the splits, the potential for annotation-driven bias is negligible.
>
> **Weakness \#5.**
> Wording consistency: To align with common usage, replace “Question and Answer (QA)” with “Question Answering (QA)” and keep this consistent.
>
> **Response.**
> Thank you for pointing this out. We have updated the wording throughout the paper to use the standard term “Question Answering (QA)” for consistency.

---

> > ### Author Response · Authors · 2025-11-25
> >
> > **Question \#1.**
> > How sensitive is ASU to the temperature parameter $\tau$?
> >
> > **Response.**
> > ASU is robust to the choice of temperature $\tau$. In Appendix H (lines 1671-1689), we report the selected temperatures across all models and datasets, and they consistently fall within a narrow range of $2.0$--$2.7$. This suggests that the optimal region is stable across settings. Moreover, Figure 2b shows that as $\tau$ increases, the negative log-likelihood rises smoothly rather than abruptly, indicating a gradual change in model behavior rather than a sharp phase transition.
> >
> > As also requested by reviewer mNw9, To further validate robustness, we include additional TOFU results on Forget05 task using $\tau \in \{2.0, 2.2, 2.4, 2.8, 3.0\}$ (with $2.3$ used in the paper Table 1). All three choices give similar forgetting and utility trends, confirming that ASU remains effective across a reasonable band of temperatures.
> >
> > |       Method    |  MU    |  FE    | Avg. |
> > |:----------------:|:-----:|:------:|:-----:|
> > | ASU$_\text{GD}$($\tau = 2.0$) | 74.21 | 75.72 | 74.97 |
> > | ASU$_\text{GD}$($\tau = 2.2$) | 72.47 | 78.04 | 75.26 |
> > | ASU$_\text{GD}$($\tau = 2.4$) | 72.06 | 79.35 | 75.70 |
> > | ASU$_\text{GD}$($\tau = 2.6$) | 71.31 | 80.98 | 76.15 |
> > | ASU$_\text{GD}$($\tau = 2.8$) | 71.38 | 81.55 | 76.46 |
> > | ASU$_\text{GD}$($\tau = 3.0$) | 71.17 | 75.00 | 73.09 |
> > | ASU$_\text{KL}$($\tau = 2.0$) | 73.88 | 75.91 | 74.89 |
> > | ASU$_\text{KL}$($\tau = 2.2$) | 72.91 | 78.04 | 75.48 |
> > | ASU$_\text{KL}$($\tau = 2.4$) | 72.34 | 79.83 | 76.08 |
> > | ASU$_\text{KL}$($\tau = 2.6$) | 71.68 | 81.05 | 76.37 |
> > | ASU$_\text{KL}$($\tau = 2.8$) | 71.31 | 81.37 | 76.34 |
> > | ASU$_\text{KL}$($\tau = 3.0$) | 71.20 | 75.71 | 73.45 |
> >
> > **Question \#2.**
> > Can you quantify the computational efficiency of ASU?
> >
> > **Response.**
> >
> > As also requested by reviewer mNw9, we report the computational cost comparison. As shown in the table below, **ASU is comparable to NPO.**
> >
> > | Method               | GPU-minutes | Memory (GB/GPU) |
> > |:--------------------:|:-----------:|:----------------:|
> > | GA$_{\text{GD}}$     | 2.36          | 44.4           |
> > | GA$_{\text{KL}}$     | 2.49          | 47.1           |
> > | NPO$_{\text{GD}}$    | 2.57          | 47.1           |
> > | NPO$_{\text{KL}}$    | 2.69          | 47.1           |
> > | DPO$_{\text{GD}}$    | 3.28          | 47.1           |
> > | DPO$_{\text{KL}}$    | 3.41          | 47.4           |
> > | IDK$_{\text{AP}}$    | 2.49          | 47.1           |
> > | IDK$_{\text{GD}}$    | 2.35          | 44.5           |
> > | IDK$_{\text{KL}}$    | 2.48          | 47.1           |
> > | ME$_{\text{GD}}$     | 2.36          | 44.5           |
> > | ME$_{\text{KL}}$     | 2.51          | 47.0           |
> > | ASU$_{\text{GD}}$    | 2.51          | 51.2           |
> > | ASU$_{\text{KL}}$    | 2.63          | 51.2           |
> >
> > This table reports GPU-minutes and peak GPU memory usage (per GPU, using 2 GPUs in total) on the forget01 task of the TOFU dataset. The results show that our method achieves a comparable computational efficiency to baselines.
> >
> > In terms of computational cost on the forget set (we do not consider the retain set, since all methods use the same loss functions, GD and KL, except for $IDK_{AP}$), GA, ME, and IDK each require one forward pass and one backward pass.
> > $$
> > Cost_{GA} \propto FLOPs_{fwd} + FLOPs_{bwd}.
> > $$
> > **ASU** and NPO require two forward passes and one backward pass.
> > $$
> > Cost_{ASU} \propto 2\,FLOPs_{fwd} + FLOPs_{bwd}.
> > $$
> > and DPO require four forward pass (two for the preferred and two for rejected responses) and one backward pass.
> > $$
> > Cost_{DPO} \propto 4\,FLOPs_{fwd} + FLOPs_{bwd}.
> > $$
> > which implies that its per-step computational cost is equivalent to NPO.
> >
> > **Question \#3.**
> > After deployment, does forgetting persist under benign continued training or repeated exposure to the forgotten content? Have you evaluated ASU’s robustness against relearning or adversarial triggering?
> >
> > **Response.**
> >
> > In this table, we report the forget efficacy of several baselines and our method on the **forget05** task in the TOFU dataset. We first unlearn the model on **forget10**, and then fine-tune it on **half of the forget set** to simulate exposure to forgotten content.
> >
> > The results show that even after relearning, ASU maintains **forget efficacy comparable to the baselines before unlearning** (reported in Table 1). Although ASU’s forget efficacy drops slightly, this is expected because the model is explicitly retrained on 50\% of the forgotten data. Overall, the results suggest that forgetting persists reasonably well under adversarial training.
> >
> > | Method          | Forget Efficacy |
> > |:---------------:|:-----------:|
> > | DPO$_\text{GD}$ | 42.51 |
> > | DPO$_\text{KL}$ | 42.63 |
> > | IDK$_\text{AP}$ | 34.58 |
> > | IDK$_\text{GD}$ | 39.39 |
> > | IDK$_\text{KL}$ | 39.21 |
> > | ASU$_\text{GD}$ | 65.61 |
> > | ASU$_\text{KL}$ | 65.67 |

---

> > > ### Author Response · Authors · 2025-11-25
> > >
> > > **Question \#4.**
> > > How resistant is ASU to jailbreak-style prompts?
> > >
> > > **Response.**
> > >
> > > To probe for forgotten information, we apply a prefix-based jailbreaking attack that prompts the model with “Sure, here is the answer:” (following [1]) and then measure forget efficacy. This setup tests how much of the suppressed content can still be recovered through prompt manipulation.
> > >
> > > The table reports forget efficacy under this jailbreak setting. ASU shows a clear advantage over all baselines, especially over refusal-based unlearning methods.
> > >
> > > | Method          | Forget01 | Forget05 | Forget10 |
> > > |:---------------:|:--------:|:--------:|:--------:|
> > > | NPO$_\text{GD}$ | 71.45 | 73.32 | 72.75 |
> > > | NPO$_\text{KL}$ | 71.88 | 73.25 | 71.14 |
> > > | DPO$_\text{GD}$ | 42.25 | 32.43 | 32.11 |
> > > | DPO$_\text{KL}$ | 42.06 | 32.45 | 32.14 |
> > > | IDK$_\text{AP}$ | 50.68 | 58.22 | 58.74 |
> > > | IDK$_\text{GD}$ | 54.77 | 50.56 | 61.72 |
> > > | IDK$_\text{KL}$ | 54.60 | 49.14 | 61.71 |
> > > | ME$_\text{GD}$  | 74.06 | 73.01 | 49.04 |
> > > | ME$_\text{KL}$  | 66.87 | 72.27 | 48.10 |
> > > | ASU$_\text{GD}$ | 85.47 | 78.55 | 78.89 |
> > > | ASU$_\text{KL}$ | 85.14 | 78.54 | 78.84 |
> > >
> > >
> > > [1] Wang, Qizhou, et al. "Towards Effective Evaluations and Comparisons for LLM Unlearning Methods." The Thirteenth International Conference on Learning Representations.

---

### Official Review · Reviewer_mNw9 · 2025-10-28

**Soundness:** 2
**Presentation:** 2
**Contribution:** 2
**Rating:** 4
**Confidence:** 3

**Summary:**

This paper proposes a computationally simple and robust unlearning framework, ASU. This framework first constructs a naturalistic forgetting teacher model by increasing the softmax temperature in the self-attention mechanism and then fine-tunes the student model to imitate the teacher on the forget set through knowledge distillation. Extensive experiments demonstrate that the proposed method outperforms existing baseline methods in both real-world and continual unlearning scenarios.

**Strengths:**

- The proposed ASU framework requires minimal architectural changes, making it highly practical for large-scale unlearning.
- The methodology is clearly described and the paper is easy to follow.

**Weaknesses:**

- As mentioned in this paper, the teacher model is constructed by applying attention smoothing, i.e., increasing the softmax temperature in the self-attention mechanism. Will this operation hurts the performance of the teacher model.
- Since this article requires a student model and a teacher model, the computational cost of forgetting should also be used as an indicator to evaluate the effect of each method.
- Can you provide a detailed proof that applying attention smoothing can make the teacher model naturally forget?
- Lack of experimental details and no mention of the model used in the paper, only mentioning Llama-2-Chat-7B.
- The effectiveness of the proposed method has not been verified on more and larger model structures.
- The selection of temperature parameters lacks theoretical guidance.

**Questions:**

How sensitive is ASU to the temperature parameter across different datasets and model scales?

---

> ### Author Response · Authors · 2025-11-25
>
> We thank the reviewer for their insightful comments. Here, we address their concerns.
>
> **Weakness \#1.**
> Since the teacher model is created by raising the softmax temperature in self-attention, it is unclear whether this modification harms the teacher model’s performance.
>
> **Response.**
> The reviewer is correct that increasing the attention temperature reduces the teacher model’s accuracy compared to the original model. This reduction is by **design**.
>
> On the forget set, the student is trained to **mimic the smoothed teacher**, intentionally weakening the model’s ability to recall factual knowledge.
>
> On the retain set and all evaluation datasets, the student is trained to remain close to the original model using KL-divergence or gradient-based retaining, which preserves utility. This setup ensures that the forget-teacher’s reduced accuracy only affects the model's knowledge about the forget set.
>
> This separation is fundamental to ASU. Attention smoothing is effective because it disproportionately disrupts factual knowledge while largely preserving general linguistic ability, enabling us to generate a forget-teacher signal that suppresses targeted facts without harming overall model quality.
>
> **Weakness \#2.**
> Because the method relies on both a teacher and a student model, the computational cost of forgetting should also be considered when comparing the effectiveness of different approaches.
>
> **Response.**
> Also, as requested by reviewer LyHx, we report the computational cost comparison. As shown in the table below, **ASU is comparable to NPO.**
>
> | Method               | GPU-minutes | Memory (GB/GPU) |
> |:--------------------:|:-----------:|:----------------:|
> | GA$_{\text{GD}}$     | 2.36          | 44.4           |
> | GA$_{\text{KL}}$     | 2.49          | 47.1           |
> | NPO$_{\text{GD}}$    | 2.57          | 47.1           |
> | NPO$_{\text{KL}}$    | 2.69          | 47.1           |
> | DPO$_{\text{GD}}$    | 3.28          | 47.1           |
> | DPO$_{\text{KL}}$    | 3.41          | 47.4           |
> | IDK$_{\text{AP}}$    | 2.49          | 47.1           |
> | IDK$_{\text{GD}}$    | 2.35          | 44.5           |
> | IDK$_{\text{KL}}$    | 2.48          | 47.1           |
> | ME$_{\text{GD}}$     | 2.36          | 44.5           |
> | ME$_{\text{KL}}$     | 2.51          | 47.0           |
> | ASU$_{\text{GD}}$    | 2.51          | 51.2           |
> | ASU$_{\text{KL}}$    | 2.63          | 51.2           |
>
> This table reports GPU-minutes and peak GPU memory usage (per GPU, using 2 GPUs in total) on the forget01 task of the TOFU dataset. The results show that our method achieves a comparable computational efficiency to baselines.
>
> In terms of computational cost on the forget set (we do not consider the retain set, since all methods use the same loss functions, GD and KL, except for $IDK_{AP}$), GA, ME, and IDK each require one forward pass and one backward pass.
> $$
> Cost_{GA} \propto FLOPs_{fwd} + FLOPs_{bwd}.
> $$
> **ASU** and NPO require two forward passes and one backward pass.
> $$
> Cost_{ASU} \propto 2\,FLOPs_{fwd} + FLOPs_{bwd}.
> $$
> and DPO require four forward pass (two for the preferred and two for rejected responses) and one backward pass.
> $$
> Cost_{DPO} \propto 4\,FLOPs_{fwd} + FLOPs_{bwd}.
> $$
> which implies that its per-step computational cost is equivalent to NPO.
>
> **Weakness \#3.**
> Can you provide a detailed proof that applying attention smoothing can make the teacher model naturally forget?
>
> Yes, we have revised the paper and provided our proof in Appendix A.
>
> **Weakness \#4.**
> The paper lacks key experimental details and does not clearly specify the models used, aside from briefly mentioning Llama-2-Chat-7B.
>
>
> **Response.**
>
> Please kindly refer to Section H (lines 1670–1689) of the appendix, where we list the exact models used for each dataset. For all baselines, we use the best hyperparameters reported in their original papers.
>
> We have also added Appendix K (lines 1782-1815), which describes our full experimental setup for the TOFU (fictitious unlearning), Real-World, and MUSE experiments.

---

> > ### Author Response · Authors · 2025-11-25
> >
> > **Weakness \#5.**
> > The method has not been tested on larger or more diverse model architectures, so its effectiveness at scale remains unverified.
> >
> > **Response.**
> >
> > We follow the experimental setup used in the standard unlearning benchmarks.
> > For the MUSE dataset [1], each task provides only a single model (LLaMA-2-7B for News and ICLM-7B for Books), so we follow the benchmark protocol.
> > In the TOFU benchmark [2], the available models are Phi-1.5-1B and LLaMA-2-7B-chat. Phi-1.5-1B has low model utility even before unlearning, making differences between baselines difficult to measure, so we focus on LLaMA-2-7B-chat.
> >
> > For the real-world unlearning setting, the benchmark in [3] includes only one model (LLaMA-3-8B) that performs well on the forget set.
> > For WMDP, we run experiments on two models: Mistral-7B and Zephyr-7B.
> >
> > Following [4], we also include additional results in Appendix L on the TOFU dataset using LLaMA-3.1-8B.
> >
> > ASU requires only a Transformer architecture, since it modifies attention weights inside the teacher and does not depend on any model-specific components beyond this. Therefore, ASU can be applied to a wide range of Transformer-based models, and the benchmarks restrict the model choices rather than the method itself.
> >
> > In addition, Training and unlearning larger models requires substantial computational power, which limits our ability to include larger architectures.
> >
> > [1] Shi, Weijia, et al. "Muse: Machine unlearning six-way evaluation for language models." arXiv preprint arXiv:2407.06460 (2024).
> >
> > [2] Maini, Pratyush, et al. "Tofu: A task of fictitious unlearning for llms." arXiv preprint arXiv:2401.06121 (2024).
> >
> > [3] Liu, Zhenhua, et al. "Learning to refuse: Towards mitigating privacy risks in llms." Proceedings of the 31st International Conference on Computational Linguistics. 2025.
> >
> > [4] Dorna, Vineeth, et al. "OpenUnlearning: Accelerating LLM Unlearning via Unified Benchmarking of Methods and Metrics." arXiv preprint arXiv:2506.12618 (2025).
> >
> > **Question \#1.**
> > How sensitive is ASU to the temperature choice when applied across different datasets and model scales?
> >
> > **Response.**
> > ASU is robust to the selection of temperature $\tau$. In Appendix H (lines 1671-1689), we report the selected temperatures across all models and datasets, and they consistently fall within a narrow range of $2.0$--$2.7$. This suggests that the optimal region is stable across settings. Moreover, Figure 2b shows that as $\tau$ increases, the negative log-likelihood rises smoothly rather than abruptly, indicating a gradual change in model behavior rather than a sharp phase transition.
> >
> > As also requested by reviewer LyHx, To further validate robustness, we include additional TOFU results on Forget05 task using $\tau \in \\{2.0, 2.2, 2.4, 2.8, 3.0\\}$ (with $2.3$ used in the paper Table 1). All three choices give similar forgetting and utility trends, confirming that ASU remains effective across a reasonable band of temperatures.
> >
> > |       Method    |  MU    |  FE    | Avg. |
> > |:----------------:|:-----:|:------:|:-----:|
> > | ASU$_\text{GD}$($\tau = 2.0$) | 74.21 | 75.72 | 74.97 |
> > | ASU$_\text{GD}$($\tau = 2.2$) | 72.47 | 78.04 | 75.26 |
> > | ASU$_\text{GD}$($\tau = 2.4$) | 72.06 | 79.35 | 75.70 |
> > | ASU$_\text{GD}$($\tau = 2.6$) | 71.31 | 80.98 | 76.15 |
> > | ASU$_\text{GD}$($\tau = 2.8$) | 71.38 | 81.55 | 76.46 |
> > | ASU$_\text{GD}$($\tau = 3.0$) | 71.17 | 75.00 | 73.09 |
> > | ASU$_\text{KL}$($\tau = 2.0$) | 73.88 | 75.91 | 74.89 |
> > | ASU$_\text{KL}$($\tau = 2.2$) | 72.91 | 78.04 | 75.48 |
> > | ASU$_\text{KL}$($\tau = 2.4$) | 72.34 | 79.83 | 76.08 |
> > | ASU$_\text{KL}$($\tau = 2.6$) | 71.68 | 81.05 | 76.37 |
> > | ASU$_\text{KL}$($\tau = 2.8$) | 71.31 | 81.37 | 76.34 |
> > | ASU$_\text{KL}$($\tau = 3.0$) | 71.20 | 75.71 | 73.45 |

---

> > > ### Comment · Reviewer_mNw9 · 2025-11-27
> > >
> > > Thank you for the author's detailed reply. Most of my concerns have been resolved, so I have increased my score.

---

> > > > ### Author Response · Authors · 2025-11-27
> > > >
> > > > We thank the reviewer for the timely response and appreciate the updated score.

---

### Official Review · Reviewer_a2Xn · 2025-10-30

**Soundness:** 2
**Presentation:** 3
**Contribution:** 2
**Rating:** 4
**Confidence:** 4

**Summary:**

The paper proposes Attention Smoothing Unlearning (ASU), a simple framework that removes memorized knowledge in large language models by flattening attention distributions through temperature scaling and self-distilling from the smoothed teacher model. ASU effectively erases factual associations while maintaining coherent and useful language behavior, outperforming existing unlearning methods across benchmarks such as TOFU, MUSE, and WMDP. This approach achieves a superior trade-off between forget efficacy and model utility, offering a practical and stable solution for real-world unlearning scenarios.

**Strengths:**

1. The classification of unlearning methods into two categories (e.g., divergence and convergence) provides an interesting conceptual framework that effectively supports the central idea of the paper.

2. The paper conducts extensive experiments on multiple benchmarks, including TOFU, MUSE, and WMDP, demonstrating the robustness of the proposed approach.

3. The paper is well organized and clearly written, making the overall argument easy to follow and the experimental results easy to interpret.

**Weaknesses:**

1. Attention Smoothing Unlearning (ASU) can be interpreted through the lens of both divergence and convergence. Specifically, emphasizing knowledge distillation aligns with convergence toward the teacher model, whereas smoothing attention may introduce divergence by disrupting previously salient attention patterns. Therefore, it remains ambiguous whether ASU should be categorized under either paradigm or considered as an independent mechanism beyond both.

2. The rationale behind how ASU mitigates the “gibberish outputs” problem is not clearly explained. Since the attention mechanism underlies linguistic understanding (e.g., grammar and syntax), smoothing attention scores may inadvertently weaken the model’s language comprehension, leading to nonsensical outputs. As shown in Figure 2, the negative log likelihood on functional knowledge does not increase as sharply as that of factual knowledge, yet it still rises beyond the default baseline.

3. The experiments are conducted on only one LLM per evaluation setting, which raises concerns about the generalizability of the proposed method across different model architectures and scales.

4. More in-depth analysis is required to substantiate the main claim. (e.g., attention score distribution after distillation, layer-wise attention smoothing)

5. It would be more practical to directly adjust the attention of the student model by treating functional and factual tokens separately, which could offer a more effective framework.

**Questions:**

See Weaknesses

---

> ### Author Response · Authors · 2025-11-25
>
> We thank the reviewer for their insightful comments. Here, we address their concerns.
>
> **Weakness \#1.**
> ASU mixes elements of convergence (through knowledge distillation) and divergence (through attention smoothing), making it unclear whether the method fits either paradigm or operates as a separate mechanism.
>
>
> **Response.**
>
> ASU follows the **convergence-based approach** by training the model toward a fixed teacher signal shaped by attention smoothing.
> We classify unlearning methods based on their **loss functions**. Divergence-based
> losses explicitly push the model away from its original behavior, while
> convergence-based losses pull the model toward a pre-defined signal. If
> “divergence’’ is defined more broadly as any disruption to the model’s original
> output probabilities or hidden states, then all unlearning methods would qualify
> as divergence-based, since they must alter the model on the forget set. The key
> difference lies in **how** this change is induced. Divergence-based losses
> (bounded or unbounded) rely on early stopping and continually decrease the model’s
> output probability on forget samples, which often harms utility. In contrast,
> convergence-based losses do not depend on early stopping and typically preserve
> utility better.
>
>
> **Weakness \#2.**
> The paper does not clarify why ASU preserves fluency, since smoothing could weaken grammar and syntax. Although functional NLL rises more slowly than factual NLL, it still increases, leaving the stability of functional language skills uncertain.
>
>
> **Response.**
> Reviewer \#pdYN also raised this question. Comparing Figure 2a and Figure 2b, we see a clear split between the factual and function tokens. For factual tokens, the NLL rises above their entropy, meaning they are effectively forgotten and unlikely to be produced. For functional tokens, the NLL stays close to or below the entropy curve, meaning that although their probabilities drop, the model can still generate them.
>
> This difference comes from how LLMs learn syntax versus facts. Syntax is learned from a huge amount of pre-training data, and functional tokens appear in many settings, so their representations are very stable. Attention smoothing weakens these associations slightly, but the effect is limited. Factual tokens, on the other hand, rely on precise associations that often come from only a few samples. When attention becomes more diffuse, these tokens are forgotten.
>
> **Weakness \#3.**
> The method is tested on only one LLM in each setting, raising doubts about how well the results generalize across different architectures and model sizes.
>
> **Response.**
>
> We follow the experimental setup used in the standard unlearning benchmarks.
> For the MUSE dataset [1], each task provides only a single model (LLaMA-2-7B for News and ICLM-7B for Books), so we follow the benchmark protocol.
> In the TOFU benchmark [2], the available models are Phi-1.5-1B and LLaMA-2-7B-chat. Phi-1.5-1B has low model utility even before unlearning, making differences between baselines difficult to measure, so we focus on LLaMA-2-7B-chat.
>
> For the real-world unlearning setting, the benchmark in [3] includes only one model (LLaMA-3-8B) that performs well on forget set.
> For WMDP, we run experiments on two models: Mistral-7B and Zephyr-7B.
>
> Following [4], we also include additional results in Appendix L on the TOFU dataset using LLaMA-3.1-8B.
>
> ASU requires only a Transformer architecture, since it modifies attention weights inside the teacher and does not depend on any model-specific components beyond this. Therefore, ASU can be applied to a wide range of Transformer-based models, and the benchmarks restrict the model choices rather than the method itself.
>
> In addition, Training and unlearning larger models requires substantial computational power, which limits our ability to include larger architectures.
>
> [1] Shi, Weijia, et al. "Muse: Machine unlearning six-way evaluation for language models." arXiv preprint arXiv:2407.06460 (2024).
>
> [2] Maini, Pratyush, et al. "Tofu: A task of fictitious unlearning for llms." arXiv preprint arXiv:2401.06121 (2024).
>
> [3] Liu, Zhenhua, et al. "Learning to refuse: Towards mitigating privacy risks in llms." Proceedings of the 31st International Conference on Computational Linguistics. 2025.
>
> [4] Dorna, Vineeth, et al. "OpenUnlearning: Accelerating LLM Unlearning via Unified Benchmarking of Methods and Metrics." arXiv preprint arXiv:2506.12618 (2025).

---

> > ### Author Response · Authors · 2025-11-25
> >
> > **Weakness \#4.**
> > More in-depth analysis is required to substantiate the main claim. (e.g., attention score distribution after distillation, layer-wise attention smoothing)
> >
> > **Response.**
> >
> > We have expanded the analysis in two ways.
> >
> > First, we added a formal proof of our main claim in **Appendix A**, which makes the link between attention smoothing, forgetting, and fluency precise.
> >
> > Second, we added new ablation studies on **layer-wise attention smoothing** (Section 5, lines 427–457), where we vary which layers are smoothed and measure the NLL and entropy on factual and functional tokens. These results give a more detailed view of how different layers contribute to forgetting factual knowledge using attention smoothing.
> >
> >
> > **Weakness \#5.**
> > It would be more practical to directly adjust the attention of the student model by treating functional and factual tokens separately, which could offer a more effective framework.
> >
> > **Response.**
> >
> > Directly adjusting the student model’s attention for factual tokens would harm overall model **utility**, because the model does not have attention heads dedicated to the forget set or the retain set. This is why the same forget-teacher can be used in the **opposite** way, which is to unlearn the retain set and preserve the forget set. which implies that the forget-teacher is **data-agnostic**.
> >
> >
> > In addition,
> > while factual and functional tokens are well-defined in the TOFU dataset, this
> > distinction does not generalize cleanly to other benchmarks such as MUSE and
> > WMDP. In these settings, tokens cannot be reliably separated into factual or
> > functional categories, making a token-specific adjustment impractical. ASU avoids
> > this issue by applying attention smoothing uniformly and letting the model’s own
> > structure determine how different token types are affected. This keeps the method
> > simple and applicable across diverse datasets.

---

### Official Review · Reviewer_pdYN · 2025-11-01

**Soundness:** 2
**Presentation:** 2
**Contribution:** 2
**Rating:** 4
**Confidence:** 4

**Summary:**

The paper proposes Attention Smoothing Unlearning (ASU), a method for removing specific knowledge from large language models (LLMs) without degrading overall performance. Instead of pushing the model away from forget data through divergence-based or entropy-maximization objectives, ASU constructs a teacher model by smoothing the attention distribution. The student model then distills knowledge from this teacher model on the forget set, effectively erasing targeted information while preserving general utility. Experiments show that ASU outperforms baseline methods across both QA and text-completion tasks, achieving stable forgetting with minimal loss of model quality.

**Strengths:**

- The paper provides a conceptually simple yet effective formulation for unlearning based on attention smoothing.
- The study includes comprehensive evaluations across QA and free-form completion settings, as well as scenario-based real-world tests.
- The experiments demonstrate that ASU consistently outperforms existing unlearning baselines.

**Weaknesses:**

- The work is heavily oriented toward experimental performance, and it lacks deeper analytical insight. The paper would benefit from additional analysis explaining why attention smoothing leads to differential effects on factual versus functional tokens.

- The paper does not provide formal guidance for selecting the optimal attention temperature parameter, leaving it heuristic.

**Questions:**

- When applying attention smoothing, why does entropy increase similarly for both factual and functional tokens, yet negative log-likelihood diverges between them, and how should this discrepancy be interpreted?

- Could the authors expand on the mechanistic explanation behind why factual information becomes more suppressed than functional information under attention smoothing, and analyze the stability of the teacher signal produced by smoothed attention? (for example, evaluating the teacher model on TOFU)

- Upon examining Table 10, it appears that the ASU-unlearned model exhibits hallucination-like behavior. This raises a concern that increasing entropy selectively for factual tokens may induce hallucinated responses, rather than safe refusal behavior. Could this mechanism therefore pose a greater risk than approaches that explicitly encourage the model to decline answering?

---

> ### Author Response · Authors · 2025-11-25
>
> We thank the reviewer for their insightful comments. Here, we address their concerns.
>
> **Weakness \#1.**
> The paper needs deeper analysis to explain why attention smoothing affects factual and functional tokens differently.
>
> Please refer to Appendix A, where we have added the proof.
>
> **Weakness \#2.**
> The paper offers no formal method for choosing the attention temperature, relying only on heuristics.
>
> **Response.**
> Appendix G (lines 1650-1670) describes our procedure for selecting the attention temperature $\tau$.
> We follow the same steps across all settings: we sweep several candidate values of $\tau$ and pick the one that gives the best forgetting behaviour while keeping the model coherent.
> Appendix H (lines 1671-1688) reports the selected hyperparameters for all datasets and models, and they are mostly
> consistent across these settings ($2.0 \le \tau \le 2.7$). These results show that the choice of $\tau$ is stable in practice.
>
> As also requested by reviewers mNw9 and LyHX, we include additional TOFU results on the Forget05 split using temperature values $\tau \in \\{2.0, 2.2, 2.4, 2.8, 3.0\\}$ (we use $\tau=2.3$ in Table 1).
>
> |       Method    |  MU    |  FE    | Avg. |
> |:----------------:|:-----:|:------:|:-----:|
> | ASU$_\text{GD}$($\tau = 2.0$) | 74.21 | 75.72 | 74.97 |
> | ASU$_\text{GD}$($\tau = 2.2$) | 72.47 | 78.04 | 75.26 |
> | ASU$_\text{GD}$($\tau = 2.4$) | 72.06 | 79.35 | 75.70 |
> | ASU$_\text{GD}$($\tau = 2.6$) | 71.31 | 80.98 | 76.15 |
> | ASU$_\text{GD}$($\tau = 2.8$) | 71.38 | 81.55 | 76.46 |
> | ASU$_\text{GD}$($\tau = 3.0$) | 71.17 | 75.00 | 73.09 |
> | ASU$_\text{KL}$($\tau = 2.0$) | 73.88 | 75.91 | 74.89 |
> | ASU$_\text{KL}$($\tau = 2.2$) | 72.91 | 78.04 | 75.48 |
> | ASU$_\text{KL}$($\tau = 2.4$) | 72.34 | 79.83 | 76.08 |
> | ASU$_\text{KL}$($\tau = 2.6$) | 71.68 | 81.05 | 76.37 |
> | ASU$_\text{KL}$($\tau = 2.8$) | 71.31 | 81.37 | 76.34 |
> | ASU$_\text{KL}$($\tau = 3.0$) | 71.20 | 75.71 | 73.45 |
>
> The table shows that temperatures in the range $2.0 \le \tau \le 3.0$ yield similar forgetting strength and model utility, indicating that ASU is stable across this range. Only at $\tau = 3.0$ we observe a drop in forgetting performance, suggesting that this temperature is slightly high and affects output coherence on the forget set.
>
> **Question \#1.**
> Why does entropy rise similarly for factual and functional tokens under attention smoothing, yet their negative log-likelihood behaves differently?
>
> **Response.**
>
> Since our default setting in the original submission smooths attention across all layers, both factual and function tokens experience a similar rise in entropy. This reflects a global increase in uncertainty. To better demonstrate the difference between factual and function tokens under attention smoothing, we ran an additional study where we smoothed only the shallow layers (e.g., layers 6–8). In this setting, both entropy and NLL for factual tokens rise much more than for function tokens. This confirms that factual tokens are more sensitive attention patterns (Please refer to Figure 4). When we use this shallow-smoothed model as the forget-teacher, we obtain nearly the same forget efficacy and model utility.
>
> Most importantly, while attention smoothing raises entropy for both token types, Figures 2a and 2b highlight a key divergence. For functional tokens, the NLL stays below or near the entropy curve, indicating that the model can still generate them. In contrast, for factual tokens, the NLL surpasses entropy, implying they will not be generated. This separation substantiates our claim that attention smoothing produces fluent but factually incorrect responses.

---

> > ### Author Response · Authors · 2025-11-25
> >
> > **Question \#2.**
> > Why does attention smoothing suppress factual information more than functional information, and how stable is the smoothed teacher signal (e.g., when evaluated on TOFU)?
> >
> > **Response.**
> >
> > We believe that this arises from the fundamentally different ways LLMs encode syntactic operations (function tokens) versus factual knowledge. Syntactic operations are learned from **massive amounts of pre-training data**, so the embeddings of functional tokens are **more stable** and resistant to attention smoothing.
> >
> > In contrast, because the specific facts to be forgotten occupy only a small portion of the pre-training corpus, the corresponding factual tokens are considerably **more sensitive** to their precise associations, which can be easily disrupted by attention smoothing. Since these associations are much less redundant than the broad patterns supporting functional tokens, smoothing weakens them quickly and suppresses the factual content.
> >
> > |       Method    |  Forget01    |  Forget05    | Forget10 |
> > |:----------------:|:-----:|:------:|:-----:|
> > | Forget Teacher($\tau = 2.0$) | 79.14 | 79.36 | 78.96 |
> > | Forget Teacher($\tau = 2.2$) | 79.04 | 79.20 | 79.70 |
> > | Forget Teacher($\tau = 2.3$) | 79.72 | 80.07 | 80.65 |
> > | Forget Teacher($\tau = 2.4$) | 80.13 | 79.32 | 79.92 |
> > | Forget Teacher($\tau = 2.6$) | 80.77 | 78.49 | 79.15 |
> > | Forget Teacher($\tau = 2.8$) | 79.83 | 77.52 | 77.59 |
> > | Forget Teacher($\tau = 3.0$) | 76.62 | 74.56 | 75.38 |
> >
> > This table shows that the forget teacher achieves strong forget efficacy on the forget set, but its performance drops when the temperature goes above 2.8 because the model starts to produce gibberish on some forget samples. But, ASU is less affected by this issue because the retain loss keeps the model from drifting toward gibberish outputs on the forget set.
> >
> > **Question \#3.**
> > Does the hallucination-like behavior in Table 10 suggest that increasing attention entropy might cause hallucinations instead of safe refusals, making ASU riskier than methods that explicitly train the model to refuse?

---

> ### Author Response · Authors · 2025-11-25
>
> **Response.**
>
> Hallucinated responses also appear in methods like **GA, NPO, and ME**, which do not train the model to refuse. In addition, we show that ASU performs better than refusal baselines on nearly all unlearning tasks when measured by forget efficacy, which reflects safety under gray-box access. This indicates that the hallucination-like behavior does not make ASU riskier than refusal-based methods, and it leads to safer unlearning.
>
> Hallucination is expected when the model is **not** directly trained to refuse.
> The goal of unlearning is to match the behavior of a retrain-from-scratch model: when queried with forget-set prompts, the retrain model usually produces coherent but incorrect answers (hallucinations) rather than
> refusals. Our attention smoothing drives the model toward coherent but incorrect answers on the forget set, which is consistent with the behavior of the retrain model.
>
> In contrast, refusal-based baselines may fail to remove the underlying factual knowledge; under adversarial attacks such as jailbreaking and relearning, they can still reveal the unwanted information. This is consistent with their lower forget efficacy in our results (e.g. Table 1). We provide additional experiments on jailbreaking and relearning at the end of this response.
>
> If, however, refusal-style outputs are desired, our method can be integrated with a refusal baseline such as $IDK_{AP}$, which would improve forget efficacy of this method while
> also generating refusal responses. It should be also considered that **refusal baselines
> are only applicable to QA format datasets, while our ASU can be applied to any dataset, such as text-completion.**
>
> In the Table below, we provided additional experiments on the TOFU dataset (under the same setup as Table 1) by combining ASU with $IDK_{AP}$. The combined approach gives better MU and FE than the baselines in Table 1. This is because ASU removes hidden knowledge that $IDK_{AP}$ alone does not forget, while still keeping the model’s outputs on the forget set as refusal responses.
>
> |       Method    | Task   | MU    |  FE    | Avg. |
> |:----------------:|:-----:|:-----:|:------:|:-----:|
> | $ASU_{GD} + IDK_{AP}$ | Forget01 | 76.67 | 80.69 | 78.68 |
> | $ASU_{GD} + IDK_{AP}$ | Forget05 | 76.15 | 83.50 | 79.82 |
> | $ASU_{GD} + IDK_{AP}$ | Forget10 | 75.60 | 86.94 | 81.27 |
> | $ASU_{KL} + IDK_{AP}$ | Forget01 | 76.75 | 80.72 | 78.74 |
> | $ASU_{KL} + IDK_{AP}$ | Forget05 | 76.24 | 83.28 | 79.76 |
> | $ASU_{KL} + IDK_{AP}$ | Forget10 | 75.61 | 86.77 | 81.19 |
>
> We have included these additional results in Table 5 as ablation study.
>
> To probe for forgotten information, we apply a prefix-based jailbreaking attack that prompts the model with “Sure, here is the answer:” (following [1]) and then measure forget efficacy. This setup tests how much of the suppressed content can still be recovered through prompt manipulation.
>
> Following table reports forget efficacy under this jailbreak setting. ASU shows a clear advantage over all baselines, especially over refusal-based unlearning methods.
>
> | Method          | Forget01 | Forget05 | Forget10 |
> |:---------------:|:--------:|:--------:|:--------:|
> | NPO$_\text{GD}$ | 71.45 | 73.32 | 72.75 |
> | NPO$_\text{KL}$ | 71.88 | 73.25 | 71.14 |
> | DPO$_\text{GD}$ | 42.25 | 32.43 | 32.11 |
> | DPO$_\text{KL}$ | 42.06 | 32.45 | 32.14 |
> | IDK$_\text{AP}$ | 50.68 | 58.22 | 58.74 |
> | IDK$_\text{GD}$ | 54.77 | 50.56 | 61.72 |
> | IDK$_\text{KL}$ | 54.60 | 49.14 | 61.71 |
> | ME$_\text{GD}$  | 74.06 | 73.01 | 49.04 |
> | ME$_\text{KL}$  | 66.87 | 72.27 | 48.10 |
> | ASU$_\text{GD}$ | 85.47 | 78.55 | 78.89 |
> | ASU$_\text{KL}$ | 85.14 | 78.54 | 78.84 |
>
>
> In the following table, we report the forget efficacy of several baselines and our method on the **forget05** task in the TOFU dataset. We first unlearn the model on **forget10**, and then fine-tune it on **half of the forget set** to simulate exposure to forgotten content.
>
> The results show that even after relearning, ASU maintains **forget efficacy comparable to the baselines before unlearning** (reported in Table 1). Although ASU’s forget efficacy drops slightly, this is expected because the model is explicitly retrained on 50\% of the forgotten data. Overall, the results suggest that forgetting persists reasonably well under adversarial training.
>
> | Method          | Forget Efficacy |
> |:---------------:|:-----------:|
> | DPO$_\text{GD}$ | 42.51 |
> | DPO$_\text{KL}$ | 42.63 |
> | IDK$_\text{AP}$ | 34.58 |
> | IDK$_\text{GD}$ | 39.39 |
> | IDK$_\text{KL}$ | 39.21 |
> | ASU$_\text{GD}$ | 65.61 |
> | ASU$_\text{KL}$ | 65.67 |
>
> [1] Wang, Qizhou, et al. "Towards Effective Evaluations and Comparisons for LLM Unlearning Methods." The Thirteenth International Conference on Learning Representations.

---

### Author Response · Authors · 2025-11-28

We thank all the reviewers for their thoughtful and constructive feedback. Below, we summarize the key points addressed in our discussion.

- **(1) Theoretical Justification [pdYN, mNw9, LyHX].**
We added a **formal proof** in **Appendix A** that clarifies how attention smoothing leads to **forgetting** while keeping the model's output **fluent**.

- **(2) Coherent Output [pdYN, a2Xn].**
We clarified in Figure 2 that function tokens have **higher NLL** than factual tokens but still remain within or **below their entropy** range, while factual tokens **exceed their entropy** and are therefore forgotten. In section 5, we clarified that, unlike function tokens learned from **massive amounts of pre-training data**, factual tokens appear **far less** in pre-training data and therefore rely on much more precise attention weights, making them easier to be forgotten.


- **(3) Stability [pdYN, mNw9, LyHX].**
We clarified that the optimal temperatures found across all datasets and models fall between **$2.0$ and $2.7$** (Appendix H). We also added new results showing that **ASU remains stable over a wide temperature range**, as reported in Table 9 of Appendix J.


- **(4) Refusal Integration [pdYN].**
We show that ASU can be **combined with refusal methods** ($\text{IDK}_{\text{AP}}$) to improve forget efficacy while keeping utility high, as shown in Table 5 in Section 5.


- **(5) Safety and Robustness [pdYN, LyHX].**
We added new adversarial experiments during discussion, including **jailbreaking** and **relearning**, showing that **ASU is more robust than the baselines** under these attacks.


- **(6) Layer-wise Smoothing Analysis [pdYN, a2Xn].**
We added ablations (Section 5) on smoothing different layers. The results (Figure 4, Table 4, and Appendix I) show that **smoothing only shallow layers is enough to forget factual content**, which matches prior work showing that factual knowledge tend to appear in early transformer layers.

- **(7) Computational Efficiency [mNw9, LyHX].**
We report GPU-minutes, memory use, and FLOP-based cost comparisons showing that **ASU has a cost similar to NPO**.

- **(8) Additional Experiments [a2Xn].**
We provided additional results on the TOFU dataset using Llama-3.1-8B (Table 10 in Appendix L).

To the best of our knowledge, this work is the **first to study how attention can be used to achieve unlearning**. This perspective lets us view unlearning as self-distillation from a forget-teacher created through attention smoothing.

---

### Meta-Review · Area_Chair_TVMA · 2026-01-06

**Summary:**

This paper proposes Attention Smoothing Unlearning (ASU), a self-distillation framework for LLM unlearning that constructs a “forget-teacher” by increasing attention softmax temperature. By flattening attention distributions, the method weakens lexical- and semantic-level associations supporting memorized facts, while retaining fluency and general utility. Experiments across TOFU, MUSE, WMDP, and real-world/continual unlearning scenarios show that ASU achieves a stable trade-off between forget efficacy and model utility, often outperforming existing baselines.

Reviewers generally found the approach conceptually simple, practical, and empirically strong. Major concerns focused on limited theoretical justification, heuristic temperature selection, the risk of hallucination-like outputs instead of refusal, robustness to adversarial prompting and relearning, and computational cost. These concerns were largely addressed during the rebuttal. The authors added a formal proof (Appendix A), layer-wise smoothing analysis, extensive temperature sensitivity studies demonstrating stability over a reasonable range, robustness evaluations under jailbreaking and relearning, and computational cost comparisons showing ASU is comparable to NPO. They also demonstrated that ASU can be combined with refusal-based methods when refusal-style behavior is desired.

Some limitations remain, including restricted model diversity due to benchmark constraints and a mechanistic explanation that is still partly empirical. Nevertheless, the method is well-motivated, the empirical evidence is strong, and the additional analyses substantially strengthen the paper. Overall, I believe this work makes a meaningful contribution to LLM unlearning and is suitable for acceptance.

**Reviewer Concerns:**

Addressed by the rebuttal: lack of theoretical justification; temperature selection and sensitivity; hallucination vs. refusal behavior.

Still partially outstanding: limited model diversity and scale; depth of mechanistic understanding.

**Reviewer Scores:**

Reviewer pdYN: Likely increased, given that most concerns were explicitly addressed and robustness analyses were added.

Reviewer a2Xn: Likely increased, as conceptual ambiguities, stability concerns, and analysis depth were substantially clarified.

Reviewer mNw9: Explicitly indicated an increased score during discussion; likely from 4 to 5.

Reviewer LyHX: Already positive.

---

### Decision · Program_Chairs · 2026-01-26

Accept (Poster)